# Better molecular preservation of organic matter in an oxic than in a sulphidic depositional environment: evidence from of *Thalassiphora pelagica* (Dinoflagellata, Eocene) cysts.

Gerard J. M. Versteegh[1,2], Alexander J. P. Houben[3,4], Karin A.F. Zonneveld[2]

[1]Heisenberg Group Marine Kerogen, Marum Research Faculty, Universität Bremen, Bremen, D-28359, Germany.
[2] Micropaleontology Group, Division Marine Palynology, Marum Research Faculty, Universität Bremen, Bremen, D-28359,
Germany.
[3]Geological Survey of the Netherlands, TNO, Utrecht, 3548 CB, The Netherlands.
[4]Marine Palynology and Palaeoceanograhy, Faculty of Geosciences, Utrecht University, Budapestlaan 4, 3584 CD Utrecht, The Netherlands.

*Correspondence to*: Gerard J. M. Versteegh (versteeg@uni-bremen.de)

**Abstract.**

Anoxic sediments as compared to oxic settings encompass a much higher proportion of relatively labile and thus more reactive organic matter, naturally giving rise to structural changes of the organic molecules themselves, as well as cross-linking between them (e.g. through reactive sulphur species).

Both processes transform the original biomolecules into geomolecules. For the oxic environment, these inter- and intramolecular transformations also operate, but cross-linking may be less important since the labile, reactive, component is rapidly removed. As such, one may expect a structurally better preservation of the more refractory initial biomolecules in the oxic environment. To test this hypothesis, initially identical biomolecules need to be compared between different preservational environments.

Here, we use the species specific morphology of organic microfossils to assure a single initial biosynthetic product (the cysts of the fossil dinoflagellate species *Thallasiphora pelagica*) for comparison. We assess the macromolecular structures of cysts from the Eocene (~40 Ma) sulphidic Rhine Graben and the oxic Kerguelen Plateau and compare them with each other and the structures of recent cysts. While between the sites the *T. pelagica* cysts are morphologically identical and show no

signs of morphological modification, pyrolysis gas chromatography mass spectroscopy and micro Fourier transform infra red analyses show that their macromolecular characteristics are strongly different. Comparison with recent cysts shows that the cysts deposited in the sulphidic Rhine Graben show a strong additional contribution of long-chain aliphatic moieties and thus less diagenetic intermolecular cross-linking. The presence of organic sulphur, identifies natural volcanisation as one of the diagenetic processes. Furthermore, we observe a loss of bound oxygen and no trace of the original carbohydrate signature of the cyst wall biomacromolecule. The material deposited in the oxic sediments of the Kerguelen Plateau shows no traces of sulphurisation. It shows a minor contribution of short carbon chains only, and thus less diagenetic intermolecular cross-linking. Furthermore, a carbohydrate signature was still preserved evidencing a better molecular preservation of the initial biomacromolecule, supporting our initial hypothesis. This shows that excellent morphological preservation doesn't imply excellent chemical preservation. It also leads to the conclusion that the best preservation of molecular structure is not necessarily where most organic matter gets preserved, which, in turn, is important for understanding the nature and fate of sedimentary organic matter and its isotopic signature.

**1 Introduction**

Kerogen, the insoluble sedimentary organic matter (OM) is by far the largest organic matter pool on Earth (Durand, 1980). It plays an important role in the biogeochemical cycles of carbon and related elements, fossil fuel formation and is a source of information on the history of life and environments. Despite considerable advances in elucidating the nature of kerogen over the last decades, many aspects of its formation, modification and structure at molecular level, are still far from understood (Vandenbroucke and Largeau, 2007; de Leeuw and Largeau, 1993; Schiffbauer et al., 2014). This is particularly true for marine kerogen where the heterogeneity and the small size of the kerogen particles are complicating factors in the effort to understand kerogen formation and modification.

Organic microfossils are a special kind of kerogen since for many of them their the specific morphology provides a direct link between the biological source and the initial biomacromolecule (often at species level) and the modified fossil geomacromolecule, for different environments and time-slices. As such analysis of their molecular structure has the great advantage that we know the initial molecular structure

so that we can separate the contributions of biosynthesis and post-mortem modification on the final kerogen structure.

The preservation of organic matter at molecular level and visible level — its apparent morphology — may be very different.

Chemical analysis of organic (micro)fossils has shown that despite excellent morphological preservation, molecular preservation may be poor, even to the extent that no recognizable trace of the original biomolecules is preserved (e.g. Stankiewicz et al., 2000; Stankiewicz et al., 1998; Gupta et al., 2007; de Leeuw et al., 2006).

At a molecular level we consider the accumulating diagenetic structural difference between the resulting geomolecule and the initial biomolecule, as an accumulating reduction of the molecular preservation. In practice, the degree of molecular preservation will be the extent to which the initial biomolecule can still be recognized by means of various analytical methods. The molecular changes may be diagenetic cross linking (kerogenisation, sensu Butterfield, 1990) but include also other kinds of structural changes, such as modifying bonds and functional groups or changing stereochemistry. These changes may take place aerobically or anaerobically and may be intramolecular (changing itself), and intermolecular (between molecules). The latter case may cause other molecules to become structural elements of the resulting geomolecule. Since this addition of structural elements can involve many different types of molecules, we prefer to use intermolecular cross linking rather than polymerization, which assumes a limited set of similar monomers. In oxic settings, organic matter degradation is mostly severe and highly selective but nevertheless processes like photodegradation and inter- and intramolecular oxidative cross-linking may lead to condensation of the OM, already early in the fossilization process which reduces OM degradability, creates macromolecules from lipids and transforms biomacromolecules into geomacromolecules (e.g. Versteegh et al., 2004; Harvey et al., 1983; Rontani, 2008; Gatellier et al., 1993). In anoxic settings OM degradation is much less complete, often leading to the formation of organic-rich sediments (e.g. Bianchi et al., 2016). Here, condensation of OM may occur in various ways, catalyzed by clay minerals, reduced metal ions, and brought about by reactive sulphur species. Of these processes the natural sulphurisation has received ample attention and this process seems especially effective in the absence of reactive Fe, which has the potential to

outcompete organic molecules for reactive polysulphides and limit organic matter sulphurization (Kohnen et al., 1989; Nissenbaum, and Kaplan, 1972; Kutuzov et al., 2019).

In comparison to anaerobic post-mortem modification of sedimentary organic matter, only little attention has been paid to the signature of aerobic processes in the geomacromolecular structure, despite the importance of aerobic modification of OM, since the Great Oxidation Event.

Most of these insights have been obtained by analyzing different taxa from unique sedimentary environments which makes it hard to obtain insight in the relation between the species specific properties of its organic product and the subsequent post-mortem modification. For such a better separation of initial organic matter and post-mortem modification ideally one would analyse the same biomacromolecule from different environmental settings.

Here we elucidate the influence of differing sedimentary environments on the chemical preservation of same species, fossil dinoflagellate cysts of *Thalassiphora pelagica*.

By far most of the cysts producing dinoflagellates are either peridinioid or gonyaulacoid. The peridinioids species are generally heterotrophic and produce brown cysts whereas the gonyaulacoids (such as cysts of *T. pelagica*) are mostly phototrophic and produce colorless cysts. Cyst degradation experiments using moorings, in surface sediments and near oxidation fronts has demonstrated that the brown peridinioid cysts rapidly degrade in oxic, and some also in suboxic environments. In contrast to this the transparent gonyaulacoid cysts are much more resistant, and often survive millennia of oxic conditions (Zonneveld et al., 2010, 2019 and references therein). These differences in degradability are also reflected in different a molecular structure of the walls of organic dinoflagellate cysts, which varies considerably between species (e.g. Mertens et al., 2016; Bogus et al., 2014; Gurdebeke et al., 2018). They all seem to consist of a carbohydrate-backbone but with various contributions of proteinaceous material that appears to be species specific (Versteegh et al., 2012; Bogus et al., 2014). Basically the same variability has been reported from fossil material where the molecular structure could even be used to distinguish between closely related *Apectodinium* species (Bogus et al., 2012). However, it became apparent that in some cases post-mortem modification may considerably change the molecular structure of the cyst wall. For instance, contributions of long-chain aliphatic moieties have been reported for *Chiropteridium* spp. (de Leeuw et al., 2006) and *T. pelagica* cysts from the Oligocene

Rhine Graben (Versteegh et al., 2007). These modifications may be so pronounced that the original signature is largely lost (de Leeuw et al., 2006; Versteegh et al., 2007). However, the extent to which environment transforms the initial biomacromolecule to its geomacromolecular derivate is still to a large extent terra incognita.

Here we concentrate on the impact of oxic versus anoxic sedimentary environments on the macromolecular structure of *T. pelagica* cysts. *T. pelagica* belongs to the transparent gonyaulacoid dinoflagellates, and thus belongs to the group of cysts resistant to aerobic degradation. This resistance enables our comparison in the first place. Characterization of the molecular composition of the cyst walls with FTIR micro-spectroscopy, py-GC-MS and thermally assisted hydrolysis and methylation (THM)-GC-MS showed that the cyst wall macromolecule of these cysts is relatively condensed which is in line with the condensation of an initially carbohydrate-based wall biomacromolecule. Nevertheless, an average chain length of ~ 12 carbon atoms was recorded which was explained by post-mortem addition of carboxylic acids by early sulphurisation of the cysts in the anoxic to suboxic setting of the depositional environment. These carboxylic acids were most probably poly-unsaturated and derived from the membranes and lipid bodies of the once living cell inside the cyst. The sulphur species readily attack functionalised groups, such as double bonds, ketones and aldehydes in organic matter (Viaravamurthy and Mopper, 1987; Schouten et al., 1994a,b; Hebting et al., 2006) and are thus well suited to crosslink unsaturated fatty acids and other functionalized components to the carbohydrate based cyst-wall macromolecule. Here we compare the results of the study on anoxically deposited and sulphurised *T. pelagica* cysts to new analysis of cysts of the same species that have been deposited in an oxic depositional environment. This allows us to determine the diagenetic effects of aerobic versus anaerobic sulphate reducing environments on the molecular structure of organic microfossils.

## 2 Material and Methods

The *T. pelagica* cysts from the oxic depositional environment (Kerguelen Plateau) have been collected during IODP Leg 120 Site 748B (58°26.45'S, 78°58.89'E) and have been isolated from sample 18H 17W 55-57 cm. The sample is derived from 152.6 m below the sea floor from the upper part of unit II consisting of nannofossil ooze with an organic carbon content < 0.2 % and a sedimentation rate of about

0.6 cm/ka. This typically open oceanic environment must have been well ventilated at the sea floor and considering the low sedimentation rate and low organic matter content the sediment has been subject to long and extensive aerobic degradation prior to reaching a burial depth below the oxic/suboxic interface. The sample has been has been deposited during the Middle Eocene Climatic Optimum, about 40 Ma ago (Schlich et al., 1989). The sample was crushed and minerals were removed using HCl (36%) and HF (40%) at room temperature. The sample was washed and the > 20 μm fraction was mounted to enable palynofacies analysis by light microscope.

The material from the Rhine Graben from the Middle Rupelian (Oligocene) has an age of about 31 Ma. At that time the Rhine Graben is a restricted and narrow basin connecting the Tethys with the North Sea. It has been deposited in a sulphate-reducing environment. The sample is derived from the Fischschiefer Unit consisting of 80 m of bituminous, finely laminated marly clay- and silt-stones typically with an organic carbon content > 3% and deposited in about 1.8 Ma, representing an average sedimentation rate of 4.4 cm/ka. The environment experienced anoxia in the water column and sediment (Pross, 2001). Freshly formed cysts sinking to the sea floor as such experienced only minimal aerobic degradation (Pross and Schmiedl, 2002). The sample was carefully cleaned and about 50 g of sediment was crushed and treated with 20% HCl to remove carbonates. Subsequently the sample was repeatedly treated ultrasonically and washed over a 125 μm sieve. The cysts were picked for FTIR micro-spectroscopy and for py-GC-MS and THM-GC-MS from the 315-125 μm fraction. HF or oxidizing agents such as $HNO_3$ or $H_2O_2$ have not been used.

## 2.1 FTIR Analysis

Preliminary analyses using transmission, reflection and attenuated total reflection (ATR) spectroscopy showed that for our material, ATR analyses provided the best results with respect to S/N ratio and scattering. Furthermore, to reduce the effects of instrument- and measurement-method-dependent differences between the spectra of the taxa compared, we re-measured cysts of *T. pelagica* (Versteegh et al., 2007) from the Rhine Graben and the culture-derived cysts of *Lingulodinium polyedrum* (Versteegh et al., 2012) using the same instrument and protocol. At least three cysts were analysed twice for each sample by micro-FTIR after cleaning the cysts with distilled water and methanol. Only

specimens were measured that were visually clean. Measurements were made with a BRUKER Invenio-S spectrometer coupled to a Hyperion 1000 IR microscope equipped with a KBr beamsplitter and liquid N$_2$-cooled MCT detector. Two hundred and fifty-six scans were obtained at 4 cm$^{-1}$ resolution using a germanium ATR crystal of 100 μm diameter. The spectra were corrected for atmospheric CO$_2$ and baseline corrected using a concave rubberband correction 64 baseline points and 5 iterations. The ATR spectra have been transformed such that the position and intensity of the absorption bands are similar to transmission spectra by using the advanced ATR correction algorithm with the an average refraction index of 1.5, a 45° angle of incidence and one ATR reflection as parameters (Nunn, and Nishikida, 2008). Assignments of characteristic IR group frequencies follow Colthup, Daly, and Wiberly (1990) and published literature. For calculation of first and second derivatives and fitting of Gaussian curves using the Levenberg-Marquardt algorithm, the software package Fytik 1.2.1 has been used (Wojdyr, 2010).

## 2.2 Pyrolysis and thermally assisted hydrolysis and methylation gas chromatography-mass spectrometry

The cysts from the Kerguelen Plateau were solvent extracted and transferred to a quartz tube (CDS 10A1-3015) for pyrolysis. For thermally assisted hydrolysis and methylation (THM) in the presence of TMAH about 0.2 ml of 10% aqueous TMAH were added and the quartz tube oven dried for 1 h at 60 °C. The sample was inserted in a CDS 5200 pyrolysis unit and kept for 5 min at 700 °C. The pyrolysate and thermochemolysate were transferred to a gas chromatograph Agilent 7890A series using a 300 °C interface and transfer-line temperature. Sample injection was in splitless mode with a delay time of 6 min. HP5MS GC column was used of 30 m length, 0.25 mm diameter and with a 0.25 μm film thickness. The GC conditions were 3 min isotherm at 40 °C followed by a temperature increase of 15 °C/min up to 130 °C, an increase of 8 °C/min to 250 °C and finally an increase of 20 °C/min to 320 °C. The final temperature was held for 5 min. The interface between GC and MS was at 280 °C. For mass spectrometry an Agilent 5975C MSD was used in full scan mode (mass range m/z 50-500 with 3.25 scans/s at 70 eV). Compound identification was performed using the NIST spectral library and relative retention times. Relative retention times were calibrated in comparison with a mixture of *n*-alkane standards run prior to each analysis.

For Pyrolysis and thermochemolysis gas chromatography-mass spectrometry of the cysts from the Rhine Graben (see also Versteegh et al., 2007) the purified cysts were transferred to a quartz tube (CDS 10A1-3015) for pyrolysis. For thermochemolysis in the presence of TMAH about 0.2 ml of 10% aqueous TMAH were added and the quartz tube oven dried for 1 h at 60 °C. The sample was inserted in a CDS AS2500-plus pyrolysis unit and kept for 5 min at 700°C. The pyrolysate and thermochemolysate were transferred to a gas chromatograph Agilent 6090N series using a 280°C inter-face temperature. Splitless mode, delay time 6 min. GC column HP5MS, 30 m length, 0.25 mm diameter, 0.25 μm film thickness. GC conditions: 3 min isotherm at 40°C, 15 °C/min to 130°C, 8°C/min to 250°C, 20°C/min to 320°C and 5 min isotherm. Interface temperature 280°C GC to MS. For mass spectrometry and Agilent 5973 MSD was used in full scan mode (mass range m/z 50-500, 3.25 scans/s, 70 eV). Prior to each analysis a mixture of *n*-alkane standards was analysed to check for contaminants and calibrate retention times.

Compound identification was performed using the NIST spectral library and relative retention times. Furthermore, we used our internal library that is based upon decades of collection of spectra with relative retention times of a wide variety of standards, natural and pyrolysis products.

## 3 Results

### 3.1 Palynofacies analysis

The assemblage consists of a nearly monotypical assemblage *T. pelagica* cysts with occasional *Phthanoperidinium* and indeterminable organic debris (Fig 1). The residue left after additional purification with a sieve with 50 μm pore-size consisted of *T. pelagica* cysts only.

### 3.2 FTIR micro-spectroscopy

The FTIR spectra are closely similar for the specimens from the same sample but there are consistent differences between the the spectra of specimens from the different samples. Therefore, only the average spectrum is shown for each sample (Fig. 2). The spectra can be divided into six absorption bands (A—F). The second derivative reveals the locations of absorptions maxima and in shoulders in much more detail (Fig. 3).

### 3.2.1 The FTIR spectra of the *T. pelagica* cysts from the oxidised depositional environment (Kerguelen Plateau)

The absorption of band A (3700—3000 cm$^{-1}$), centred at 3400 cm$^{-1}$ is assigned to alcoholic OH, phenolic OH, and/or carboxylic OH.

The strong, narrow aliphatic absorptions of band B (3000 cm$^{-1}$—2700 cm$^{-1}$), centred at 2928 and 2863 cm$^{-1}$ are assigned to antisymmetric stretching vibrations from $CH_2$ and symmetric stretching vibrations from $CH_2$ methylene groups, respectively. Further deconvolution shows that a third Gaussian/Lorentzian band at 2892 cm$^{-1}$ is required to model this part of the spectrum (Fig 4). This third band is attributed to CH stretch. There is a marked absence of evidence for absorptions by $CH_3$. In algal walls such a low contribution of methyl groups is typically associated with the presence of long aliphatic chains in the wall macromolecule, such as the algaenan of e.g. *Chlorella emersonii* (Fig 2). However, the corresponding absorption of $CH_2$ in long carbon chains near 723 cm$^{-1}$ (see also McMurry, and Thornton, 1952) is also absent from our spectra leading to the conclusion that the *T. pelagica* wall macromolecule consists primarily of short carbon chains without free methyl groups, which is in agreement with the pyrolysis and spectroscopy based hypothesis that the gonyaulacoid dinoflagellate cyst walls (such as e.g. *L. polyedrum* ) primarily consist of a carbohydrate-based biopolymer (Versteegh et al., 2012).

For band C (1800 — 1500 cm$^{-1}$) the absorption centred at 1705 cm$^{-1}$ as well as the shoulder 1768 cm$^{-1}$ we assign to the vibration of carbonyl C=O. Hereby, the absorption at 1705 cm$^{-1}$ may represent α and β unsaturated aliphatic ketones or hydrogen bonding in carboxylic acid dimers (Colthup et al., 1990) the corresponding -enol tautomers may provide a second explanation for the moderate absorption centred at 1612 cm$^{-1}$.

For band D (1500—1300 cm$^{-1}$), the absorption at 1411 cm$^{-1}$ may be attributed to deformation of $CH_2$ next to carbonyls. Alternatively the absorption at 1411 cm$^{-1}$ may arise from olefin CH rocking. Absorptions at 1463 cm$^{-1}$ we relate to scissors deformation of $CH_2$. There is no clear symmetric umbrella deformation of C-$CH_3$ at 1375 cm$^{-1}$ of hydrocarbons, which agrees with the absence of evidence for methyl groups in band B.

For bands E to G, the weakly defined absorptions between 1305 and 1203 cm$^{-1}$ (Band E; 1300—1150 cm$^{-1}$) can be related to C-O carboxylic acids. The absorptions in band F (1150—900 cm$^{-1}$) at 1112, 1047, and 986 cm$^{-1}$ are assigned to C-O e.g. as C-O-C and C-O-H of carbohydrates. For band G the minor absorption at 728 cm$^{-1}$ is attributable to CH$_2$ rock, its low intensity agrees with only a minor contribution of predominantly short carbon chains < 4 CH$_2$ sequences (McMurry, and Thornton, 1952). As such the spectrum indicates a relatively short-chain polymer. The absence of evidence for CH$_3$ indicates it to be heavily branched and/or functionalised. The richness in C-O, and O-H suggests the functionalities to be primarily hydroxyl groups and ether bonds and a much lower abundance of carbonyl groups; a combination typically found in carbohydrate-based polymers. There is no evidence for an aromatic contribution but some aliphatic C=C can't be excluded.

### 3.2.2 The FTIR spectra of re-analysed cultured *L. polyedrum* cysts and *T. pelagica* cysts from the Rhine Graben

The new spectra for *L. polyedrum* differ from the published spectrum (Versteegh et al., 2012). We measured in ATR mode and the spectrum published has been obtained in reflection mode. Re-measuring also in reflection mode, the spectrum was very similar to the published one (Supplementary information figure S1), confirming that the observed differences result from a different mode of measurement and spectrum correction. The much larger relative abundance of bands A and B and reduction of band F result from a wavenumber dependent amplification of the absorption of the ATR correction algorithm. As such the differences between the published spectra and those presented here, predominantly result from different measurement and spectrum correction algorithms and do not require a new spectral interpretation. Since Mie scattering causing spectrum distortion plays a relatively large role for these small objects in reflection mode (Herman 1962), we use the corrected ATR spectra for comparison.

The new ATR spectra obtained for *T. pelagica* from the Rhine basin are closely similar to the transmission spectrum published except for band F (Supplementary information figure S2), which was much weaker in the spectra of Versteegh et al. (2007). This difference is unlikely to be accounted for by degradation of the sample since over the years it has been kept frozen at -20°C in distilled water. Also here we attribute the observed difference to the different modes of measurement (ATR versus

transmission) and the difficulty of measuring single cysts in transmission mode. The interpretation of the absorption bands is similar to that for the specimens from the Kerguelen plateau but there are large differences in relative strengths of these bands, notably the stronger relative absorptions of B, C and to some extent D.

### 3.3 Pyrolysis–GC-MS

The pyrolysate (Fig.5; Table 1) contains both aliphatic and aromatic moieties. There is a considerable number of oxygen containing saturated and unsaturated non-aromatic moieties with 5-membered and 6-membered rings and up to three additional carbon atoms amongst which are furans, cyclopente/anones, phenols, benzofurans, indanones and naphthalenones and naphthalenols. Furthermore, there are several series of aromatic moieties. Phenol (*m/z* 94) and methylated phenols (*m/z* $94 + 14 * n$) have been detected up to $n = 4$. Benzene (*m/z* 78) and alkyl benzenes (*m/z* $78 + 14 * n$) have been detected up to $n = 9$. Compounds with more than three carbon atoms attached to naphthalene and indene were not detected. Linear carbon chains play only a minor role and their distribution becomes only apparent from mass chromatograms of their characteristic ions. *n*-Alkanes and n-alkenes (*m/z* 83 and 85) show alkane/alkene pairs for $C_7$ - $C_9$ and a series of only n-alkanes for $C_{14}$ - $C_{21}$. The mass chromatogram *m/z* 58 reveals a series of *n*-alkan-2-one / *n*-alken-2-one pairs from $C_6$ - $C_{12}$ and *m/z* 60 shows $C_3$ - $C_{10}$ alkanoic acids.

The mass distribution in the first 28 min of the chromatogram (Fig. 6) shows that the masses representing aromatic moieties (such as *m/z* 91, 105, 115, 117, 119, 128, 129) clearly emerge above their neighbour masses (*m/z* $\pm$ 5). The masses representing saturated aliphatic moieties (*m/z* 57, 71, 85) do not emerge above background as is the case for alkenes $> C_5$ (*m/z* 83, 97).

### 3.3 THM–GC-MS

THM-GC-MS (Fig.7; Table 2) clearly shows methyl esters of $C_3$—$C_9$ and $C_{14}$—$C_{16}$ *n*-alkanoic acids ($C_{17}$ and $C_{18}$ could not be detected) and $C_4$—$C_9$ $\alpha,\omega$ dicarboxylic-dimethylesters the latter with a strong preference for the butanedioic moiety. Few oxygen containing aromatic compounds are present. The

most prominent are mono- to tri-methoxy benzenes and mono-and dimethoxy benzoic-acid methylesters. Traces of alkane/alkene doublets (using *m/z* 83/85) are absent.

## 4 Discussion

### 4.1 Palaeoenvironment

The Rhine Graben restricted marine sedimentary environment is very different from that of the open ocean Kerguelen Plateau. The Rhine Graben sediments (Pross, 2001) are about 10 Ma younger, the apparent sedimentation rate (4.4 cm/ka) and total organic carbon contents (>3%) are at least an order of magnitude higher than at the Kerguelen Plateau (sedimentation rate 0.6 cm/ka; and TOC < 0.2%, Schlich et al., 1989). However, the most important factor for organic matter diagenesis and preservation is the redox environment, notably the oxygen exposure time of the organic matter. With the change from an aerobic redox environment via progressively less powerful electron acceptors to sulphidic conditions the amount of energy gained during the oxidation of organic matter also progressively decreases, and OM degradation becomes increasingly thermodynamically limited (Arndt et al., 2013). Therefore, in settings with short oxygen exposure not only the TOC contents, but also the concentration of more labile and reactive organic matter remains much higher over geological time in comparison to equivalent settings with longer oxygen exposure (e.g. Hedges and Keil, 1995, Hoefs et al., 2002, Versteegh et al., 2010, Nierop et al., 2017). Thus, the Rhine Graben sediments with anoxia already in the water column and sulphidic conditions in the sediment never experienced prolonged aerobic degradation. As such sedimentary organic carbon concentrations are higher as is the reactivity of this organic matter in the presence of reactive sulphur species. The Kerguelen sediments experienced much longer oxygen exposure. This exposure was long enough to enable demineralization of most of the OM reaching the sea floor, resulting in sediments lean in OM and where this OM is refractory. With sedimentation continuing, the sediments containing the *T. pelagica* cysts will have become suboxic over time. However, the refractory nature of the OM and its low concentration severely limit anaerobic intermolecular cross-linking, due to a lack of available substrate. Furthermore, OM mineralization rates will have remained so low that the system never reached sulphate reduction and release of reactive

sulphur species. This implies the OM surviving until the environment becomes suboxic, may be
expected to remain diagenetically relatively unaltered from that moment.

## 4.2 Comparison of the FTIR spectra (Fig. 2)

Infrared analysis of cysts from surface sediments and dinoflagellate cultures shows that the cysts cluster
in two groups (Bogus et al., 2014; Ellegaard et al., 2013). (1) Transparent cysts with a cellulose-like
cyst wall such as *Lingulodinium polyedrum* (Versteegh et al., 2012), and (2) Brown cysts with a chitin-
like cyst wall such as *Polykrikos kofoidii* (Bogus et al., 2014). The former resist aerobic degradation, the
readily degrade aerobically. The extinct genus *Thalassiphora* produces cysts that belong to the first,
degradation resistant group as is apparent from their morphology. This also explains their survival in the
Kerguelen Plateau sediments.

The spectra of the *Thalassiphora* cysts from the Kerguelen Plateau are very similar to those of the
culture-derived *Lingulodinium* cysts (Fig. 2A) except for the considerably lower absorptions for the E
and F bands (various C-O stretching modes). This similarity implies only little change to the cyst
macromolecule.

Since this is not accompanied by a loss in band A (O-H) we attribute this difference primarily to a
relative loss in ether bonds rather than hydroxyl groups. In the Kerguelen Plateau setting the relative
decrease in ether bonds is likely to be achieved through oxidation, of the carbohydrate. This leads to
ring opening, chain-breaking and increases the relative abundance of carboxylic acids, aldehydes and
ketones which, in turn may increase the intramolecular and intermolecular hydrogen bonding (Potthast
et al., 2006) and thus reduces the relative abundance of ether bonds. Additionally, oxidative cross-
linking may add carboxylic acids to the cyst wall, a process already starting before burial (Blom, 1936;
Harvey et al., 1983; Versteegh et al., 2004; Stankiewicz et al., 2000). Plankton is rich in unsaturated
fatty acids which will reach the sea floor and play a role in early diagenesis. Oxidative cross-linking,
attacks primarily the double bonds of unsaturated fatty acids, leading to fragmentation of the fatty acids
as well a cross-linking. This addition will increase in O-H and C=O relative to $CH_2$ (Lazzari and
Chiantore, 1999; Scalarone et al., 2001). The carboxylic groups are less reactive and remain largely

unaltered during this process (Scalarone et al., 2001). This condensation process thus also adds C=O, O-H and $CH_2$ and reduces the relative proportion of ether bonds in the resulting cyst geomacromolecule.

The spectra of the *Thalassiphora* cysts from the Rhine Graben differ much from the spectra of culture-derived *Lingulodinium* cysts than the cysts from the Kerguelen Plateau (Fig. 2B). Compared to the latter, there is significantly less absorption in bands A (O-H), more in band B ($CH_x$) (Fig 2C) whereas in

band D the broad maximum at 1407 $cm^{-1}$ splits giving rise to maxima at 1448 ($CH_2$ deformation and $CH_3$ asymmetric deformation) and 1376 $cm^{-1}$ ($CH_3$ deformation). In contrast to cysts of *T. pelagica* from the Kerguelen plateau the cysts from the Rhine Graben have been deposited in an environment with sulphidic conditions (Versteegh et al., 2007); sedimentation rate and the sedimentary OM content are an order of magnitude higher (Pross and Schmiedl, 2002). In this sulphidic environment with ample

labile OM the most easily explanation is the addition of longer-chain aliphatics such as carboxylic acids to the cyst wall through anaerobic cross-linking such as natural vulcanisation. Reactive sulphur species particularly attack aldehydes, ketones, conjugated, distal and to a lesser extent mid chain C=C double bonds but react much less with carboxyl and hydroxyl groups (Viaravamurthy and Mopper, 1987; Schouten et al., 1994a,b; Hebting et al., 2006). The strong decrease in O-H absorption and increase in

C=O seem to disagree with this. However, the initial carbohydrate-based cyst wall is intrinsically poor in O-H and C=O, chemical modification such as addition of alkanoic acids automatically decreases the relative abundance of these bond types and increases $CH_x$ and C=O. Moreover, in this anaerobic environment oxygen as a powerful electron acceptor is in high demand and will selectively be removed, contributing to the loss of hydroxyl groups of the cyst macromolecule. Also here, the amply available

polyunsaturated carboxylic acids from the marine plankton including the dinoflagellate cysts of which many died without hatching (Pross, 2001) provide a rich source of lipids for this process, increasing the relative abundances of foremost $CH_2$ and C=O relative to C-O and O-H. Available clay minerals and free metal ions may have played a catalytic role (e.g. Schiffbauer et al., 2014) primarily outside the cyst wall.

As discussed above, both the Kerguelen Plateau and the Rhine Graben sedimentary environments have modified the *T. pelagica* cysts, albeit in different ways. Infra-red analysis of extant cysts of other members of this group of Gonyaulacoid dinoflagellate cysts show structural differences between the

species (Versteegh et al., 2012; Mertens et al., 2016; Mertens et al., 2015; Mertens et al., 2015; Bogus et al., 2014; Gurdebeke et al., 2018; Luo et al., 2018). Hence, the question arises to which part of the differences recorded between the spectra of these *T. pelagica* with other Gonyaulacoid taxa arises from differences in cyst biosynthesis and what part arises from post-mortem modification. The spectra of the recent Gonyaulacoid cysts show a consistent pattern in their order of absorption strength for bands D, E and F with band F being strongest and band E weakest. This pattern is also seen for the cysts from the Kerguelen Plateau. However, for band C this is more complicated. Analysis of recent cysts by Bogus et al. (2014) shows for all members of the Gonyaulacaceae (*Lingulodinium, Spiniferites, Impagidinium, Protoceratium*) a band C with weaker absorption than bands D and E). In Gurdebeke et al. (2018), all members of the gonyaulacacean *Spiniferites* (except *S. pachydermus* which spectrum was taken from Bogus et al., 2014) show a strong absorption near 1600 cm$^{-1}$ in band C that exceeds the absorptions in band D. This we also see for *T. pelagica* which, considering the variability of this band in recent cysts, we consider intrinsic feature of the biopolymer. In band C recent gonyaulacacean cysts also show a shoulder near 1705 cm$^{-1}$, which is always of less amplitude than the absorption 1590 cm$^{-1}$. This is clearly different for *T. pelagica* for which the absorption at 1705 cm$^{-1}$ is the most pronounced one in band C and for this difference with recent cysts we attribute the high absorption at 1705 cm$^{-1}$ for *T. pelagica* to diagenesis.

To get better insight in the role of diagenesis we have to directly compare the cysts from the Rhine Graben and the Kerguelen Plateau (Fig. 2C). They represent the same species *T. pelagica*. Therefore, we may assume identical or at least closely similar chemical structures prior to diagenesis so that all spectral differences between them logically reflect differences in preservation. Clearly, this only provides the accumulated differences, changes common to both the cysts from the Kerguelen Plateau and the Rhine Graben are not apparent. The differences lie foremost in the higher contribution of band A (O-H) and lower of band B (CH$_x$) for the cysts from the Kerguelen plateau. The spectrum of these cysts from the Kerguelen Plateau also more closely matches the spectrum of *L. polyedrum.* Indeed, with the stronger absorption by CH$_x$ and relatively weak absorptions of the other bands the spectrum of the Rhine Graben specimens has become much more similar to that of the algaenan walls of green algae such as *C. emersonii* (Fig. 2D; Allard, and Templier, 2001) or *Tetraedron minimum* (Blokker et al.,

1998; Goth et al., 1988) albeit without the absorption at 718 cm$^{-1}$ and thus lacking the long carbon chains of algaenan.

As such it appears on the basis of infrared spectroscopy of both *Thalassiphora* samples in comparison to recent cysts from cultures and surface sediments that we must assume that part of the observed differences with the *Lingulodinium* cysts may result from differences in cyst biosynthesis. The infrared spectra of recent *L. polyedrum* and fossil *T. pelagica* from the Kerguelen Plateau are very similar suggesting that the cysts of *L. polyedrum* are a good modern analogue. Nevertheless, the stronger absorption for the fossil cysts of probably conjugated carbonyl at 1705 cm$^{-1}$ can be assigned to diagenesis. Further comparison with *Lingulodinium* shows that cysts from the Kerguelen Plateau are depleted in ether bonds. This depletion we attribute to modification of the cyst wall by oxidation and oxidative cross-linking. The cysts from the Rhine Graben differ considerably more from the *L. polyedrum* cysts than those from the Kerguelen Plateau. These differences are particularly an enrichment in CH$_x$ and to a lesser extent C=O, and show a relative loss of O-H. These differences are attributed to reactions by reactive sulphur species crosslinking the cyst wall and adding linear carbon chains to it. Furthermore, in the anaerobic depositional environment of the Rhine Graben, we expect cyst diagenesis to lead to a loss of oxygen, rather than an increase as we observe for the cysts from Kerguelen Plateau.

### 4.3 py-GC-MS and THM-GC-MS

GC-MS results are consistent with the results of the infrared analyses, which suggest that differences in depositional environment are associated with differences in molecular structure. Pyrolysis of macromolecules consisting of long alkyl chains, such as algaenans or polyethylene, typically produces a series of alkane/alkene doublets (Fig. 5). The presence of such *n*-alkane/alkene doublets up to C$_9$ (and similarly C$_7$—C$_{11}$ methyl-alkanone/alkenone doublets) in the cysts from the Kerguelen Plateau suggests that these have been derived from the cyst macromolecule. The length, up to C$_9$ may not be accidental since many unsaturated *n*-alkanoic acids have double bonds at $\omega$-9 or closer to the end of the molecule and oxidative cross-linking operates at such double bonds whereby the remainder of the molecule may be released (e.g. Lazzari and Chiantone, 1999). Upon pyrolysis, the chains will break at these cross-

links resulting in a characteristic chain length distribution. The absence of alkane/alkene doublets from $C_{10}$ in these cysts suggests that the aliphatic chains were relatively short which agrees with the absence of an absorption near 720 cm$^{-1}$ in the infrared spectrum. The absence of $> C_{10}$ *n*-alkanoic acids supports this further. The presence of a minor abundance of *n*-alkanes without accompanying *n*-alkenes for $C_{14}$—$C_{21}$ and their absence for $C_{10}$—$C_{13}$ suggests that these $C_{14}$—$C_{21}$ *n*-alkanes either were not covalently bound to the cyst macromolecule and they may have been absorbed by the cyst wall and failed to be removed upon extraction or were exclusively ester-bound so that upon their release no *n*-alkenes were formed.

For the Kerguelen Plateau cysts we find a similar gap in distribution in the *n*-alkanoic methyl-esters released upon and methylated by the thermochemolysis with $C_{10}$—$C_{13}$ absent, $C_{14}$—$C_{16}$ present, but no longer chains (Fig. 6). The absence of the ubiquitous $C_{18}$ *n*-alkanoic methyl ester indicates that these longer $C_{14}$—$C_{16}$ chains are not simply contamination but that also these were associated to the cysts themselves. In dinoflagellates, the saturated carboxylic acids are often dominated by $C_{16}$ with $C_{14}$ second in abundance (e.g. Mansour et al., 1999; Robinson et al., 1987; Mooney et a., 2007; Chu et al., 2009) and this same pattern obtained by thermochemolysis supports the hypothesis that these lipids originate from the nearest source, the dinoflagellate cytoplasm. The shorter $C_3$—$C_9$ *n*-alkenoic methyl esters are accompanied by ($C_4$ — $C_9$) alkanedioic dimethyl esters with a strong dominance of the $C_4$ representative and, provide evidence for a cyst wall with short-chain ester-cross-linked moieties. Fresh *L. polyedrum* cysts do not show any aliphatic signature so that we attribute the aliphatic signature in the *T. pelagica* cyst wall to diagenetic modification of the initial macromolecule.

The thermochemolysis of cultured *L. polyedrum* cysts produced a series of mono- to penta-methoxy-benzenes, evidencing the carbohydrate hydroxy groups in the cyst-wall polymer (Versteegh et al., 2012). We observe up to tri-methoxy-benzenes as thermochemolysis products of the cysts from the Kerguelen Plateau, but they are of much less abundance. This suggests that the carbohydrate nature is still recognizable in these fossil cysts. In contrast to this, di-, to penta-methoxy-benzenes have not been detected as thermochemolysis products of the cysts from the Rhine Basin indicating that for those cysts the carbohydrate nature has been entirely lost.

65  For the *T. pelagica* cysts from Kerguelen Plateau, the detection of benzenes with up to nine attached carbon atoms and phenols with up to four attached carbon atoms fully agrees with the observed chain length distribution of the aliphatic component and again suggests a macromolecule with carbon chains shorter than $C_9$. These series are longer than for the cultured *L. polyedrum* cysts but shorter than for the *T. pelagica* cysts from the Rhine Graben.

70  The stronger diagenetic alteration and sulphurisation of the material from the Rhine Graben is also reflected by the relatively higher abundance of masses characteristic for sulphur (*m/z* 64, $SO_2$) and alkylthiophenes (*m/z* 97, 111, 125 and 139 together with a contribution by longer *n*-alkenes) whereas oxygen containing moieties are more abundant in the material from the Kerguelen Plateau (*m/z* 93, 107, 121, 135 from phenols and *m/z* 95, 145 from (benzo)furans) as are alkylbenzenes (*m/z* 79, 105, 119,

75  133) (Fig. 8). Comparison to mass distributions of pyrolysates of other macromolecules (Fig. 6) demonstrates the much lower aliphatic contribution of the Kerguelen material, and the disappearance of oxygen from, and addition of sulphur to the cysts from the Rhine Graben.

**4.4 Cyst wall diagenesis, diagenetic processes and implications**

The diagenetic modification of the cyst walls is complex and involves different processes at different

80  times and in different environments. We do not pretend that we can resolve all these processes and modifications. What we can identify is the presence of sulphurisation, the presence of linear carbon chains and the extent to which the resulting cyst geomolecule deviates from the modern analogue biomolecule.

As a result of the very different sedimentary environments of the Rhine Graben and the Kerguelen

85  Plateau the initially identical cyst biomacromolecules of the *T. pelagica* cysts have been transformed into clearly different geomacromolecules. The cyst geomolecule from the Kerguelen Plateau contains still a lot of oxygen-bearing functionalities and only a small amount of predominantly short linear carbon chains have been detected. Upon thermochemolysis, the carbohydrate nature of the wall material still can be evidenced. This is explained by the paleoenvironmental setting with oligotrophic surface

90  waters, the aerobic depositional environment and low sedimentation rates. Here, oxic degradation and oxidative cross-linking prevail during transport of organic matter through the water column and in the

surface sediments. We consider these processes to be most important modifiers of the cyst macromolecule for the Kerguelen Plateau environment. In such an environment with considerable oxygen exposure times all but the most refractory organic matter has been rapidly remineralized aerobically resulting in organic carbon lean sediments and only little substrate to contribute to the cyst walls through oxidative cross-linking. In contrast to this, anoxia prevailed in the water column and sediment in the Rhine Graben environment. The Rhine Graben sediments remained rich in lipids and more refractory organic matter (Böcker et al., 2017) due to thermodynamic limitation of OM degradation under anaerobic (compared to aerobic) conditions (e.g. Arndt et al., 2013). As a result, there is ample substrate to react with the cyst biomacromolecule, which explains the ample presence of linear carbon chains (up to $C_{20}$) associated to the cyst wall. The presence of alkyl thiophenes upon in the pyrolysate identifies natural sulphurisation as one of the mechanisms turning the cyst biomacromolecule into a geomacromolecule. As a result of the diagenetic modification, the cysts macromolecule shows an overall loss of oxygen bearing functional groups and its carbohydrate signature can't evidenced anymore upon thermochemolysis. On the basis of the results from pyrolysis, thermochemolysis and infrared analyses it appears thus that the cysts from the Kerguelen Plateau structurally more closely resemble the recent analogue than the cysts from the Rhine Graben and as such display better structural preservation. The cysts from the Rhine Graben not only display stronger intermolecular cross linking but structural changes to the initial molecule also resulted in loss of its the carbohydrate signature upon thermochemolysis.

Experiments show that both the oxidative cross-linking (Lazzari and Chiantore, 1999; Scalarone et al., 2001) and sulphurisation (Schouten et al., 1994a,b) preferably attack double bonds. Subsequently the activated group cross-link with a nearby carbon atom or functional group of the same (macro)molecule or a different one. For oxidative cross-linking the aliphatic chain may break at the double bond position during this process, which could provide an additional explanation for the absence of long carbon chains in the pyrolysate of the cysts from the Kerguelen Plateau. As far as we are aware of, this shortening mechanism has not been observed for natural sulphurisation. Furthermore, and not surprisingly, the aerobically modified cysts appear to be rich in oxygen bearing functionalities whereas a general loss of such functionalities is observed for the cysts from the anaerobic setting. This

difference in organically bound oxygen provokes the speculation that repeatedly changing organic matter from oxic to anoxic states such as occurs in bioturbated sediments effectively supplies the anoxic environment with organically bound oxygen facilitating anaerobic organic matter degradation whereas the change from anoxic to oxic conditions enables aerobic processes to oxidise labile functionalities created anaerobically. This could be an additional explanation why redox oscillation promotes OM

remineralisation (Aller, 2001).

For the Kerguelen Plateau, the aerobic degradation will rapidly decrease the OM concentration, foremost of the most reactive and labile components. As such, the rate of modification of the cyst molecular structure as a result of interaction with other OM will rapidly decrease. Due to ongoing sedimentation, the cysts will get out of the reach of oxygen diffusing into the sediment and enter the

suboxic zone. The low concentration and refractory nature of the remaining OM will have allowed for low suboxic diagenetic rates too and the system will never have reached the sulphate reduction zone. Therefore we consider that for the cysts from the Kerguelen Plateau, by far most molecular change will have been in an early, aerobic stage, especially with respect to interaction with other organic matter.

In contrast to this, the Rhine Graben sediments have an order of magnitude higher OM contents,

whereas the anaerobic redox environment promotes the production and accumulation of small, relatively labile organic molecules and reactive sulphur species. Initially, with a lot of fresh OM, mineralisation rates were similar to aerobic environments (Arndt et al., 2013). We suggest that this also accounts for the diagenetic modification of the cyst molecule. In contrast to the Kerguelen Plateau, the high OM content must have supported active anaerobic degradation and with it the generation of

reactive organic and inorganic molecules for a much longer time. Therefore, it seems logical that the accumulated molecular change (through intermolecular cross-linking, and structural modification) of the cysts from the Rhine Graben is larger than for the cysts from the Kerguelen Plateau.

Finally, despite the changes taking place to the *T. pelagica* cysts at molecular level, at microscopic level nothing of these changes is apparent. Therefore, excellent morphologic preservation is no proof of

excellent molecular preservation and this should be kept in mind e.g. upon isotopic analyses of organic matter. Fortunately, a simple infrared analysis, at least in our example, may be sufficient to identify to what extent chemical modification may have taken place. Finally, the nature of the cysts is such that

they belong intrinsically to the refractory organic matter so that the changes brought about in the anaerobic environment do not introduce bias du to a higher presence of these cysts in the fossil record.

## 5 Conclusions

By analysing remains of the same species (*T. pelagia*) from two contrasting environmental settings, differences in the composition of the starting material can be excluded. As such, all chemical differences between the cysts from the different depositional settings can be assigned to post-mortem modification of the original biomacromolecule. However, alteration processes that the different settings have in common will remain undetected and requires knowledge of the initial state. Here we circumvent the absence of such material by comparing to a close analogue, recent cysts of *L. polyedrum.*

Analysis of the composition and structure of *T. pelagica* cysts from the Kerguelen Plateau and comparison to recent cysts of gonyaulacoid dinoflagellates, the group of which *Thalassiphora* is a member, shows that the fossil cysts to a large extent have been structurally preserved. Diagenetic modification is mainly confined to an increase in C=O a loss of C-O and presumably an increase in hydrogen bridges. Thermochemolysis generates despite a strong decrease in their production di- and tri-methoxybenszenes indicating that part of the carbohydrates nature has preserved. The presence of carbon chains up to $C_{11}$ is also attributed to diagenesis since such an aliphatic component is absent in culture-derived gonyaulacoid cysts.

Comparison with modern gonyaulacoid cyst walls demonstrates that the *T. pelagica* cysts from the Kerguelen Plateau and Rhine Graben have been structurally changed by diagenetic intermolecular crosslinking and by intermolecular modification. Both processes were stronger for cysts from the Rhine Graben than the 10 Ma younger specimens of the same species from the Kerguelen Plateau and despite the absence of any visual indication for these differences upon microscopic examination. In contrast to the cysts from the Kerguelen Plateau, the cysts from the Rhine Graben lost their carbohydrate signature of the initial biomacromolecule. We attribute this to a stronger diagenetic change enforced by the organic rich and sulphidic diagenetic setting the Rhine Graben material has been subject to. This study demonstrates the advantage of using organic particles of narrowly defined (species level) biological sources for unravelling post-mortem modification of organic matter. It also shows that excellent

morphological preservation doesn't imply excellent structural preservation at molecular level and that the post-mortem modification of the same initial biomacromolecule can differ considerably between sedimentary settings. Furthermore, it leads to the counterintuitive conclusion that the best preservation of molecular structure may not be found where most organic matter is preserved.

## 5 Code and Data Availability

Data are available at https://doi.pangaea.de/10.1594/PANGAEA.905696

## 6 Author Contributions

Gerard J. M. Versteegh collected the geochemical data interpreted them and wrote the manuscript. Alexander J. P. Houben provided the material and did the palynology. Karin A. F. Zonneveld was responsible for project funding and administration and together with GJMV formulated the overarching research goals and aims. All authors discussed data and earlier manuscript versions.

## 7 Competing Interests

The authors declare they have no competing interests

## 8 Acknowledgments

We thank Henk Brinkhuis (Utrecht University, Royal NIOZ) for his constructive comments on the manuscript. We also thank reviewers Morgan Raven and Jacob Vinther for their constructive comments, which significantly improved the manuscript. This work has benefited from funding by the German Science Foundation (DFG) Grant MER/MET: 17-87. IODP is thanked for providing the sample utilized in this study.

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

**Table 1. Compounds identified in the pyrolysate[a]**

| # | Name | M+ |
|---|------|-----|
| 1 | Methylcyclopentadiene ($C_6H_8$) | 80 |
| 2 | Benzene | 78 |
| 3 | Cyclohexadiene | 80 |
| 4 | 2,5 Dimethylfuran | 96 |
| 5 | Toluene | 92 |
| 6 | ? $C_8H_{12}$ | 108 |
| 7 | Cyclopentenone + Furfural | 96 + 82 |
| 8 | Ethyl-benzene | 106 |
| 9 | 2-Propylfuran | 110 |
| 10 | 1,3 + 1,4 dimethylbenzene | 106 |
| 11 | Pentanoic acid | 102 |
| 12 | 1,2 dimethylbenzene | 106 |
| 13 | Styrene | 104 |
| 14 | 2-Methyl-2-cyclopentenone (67, 96) | 67, 96 |
| 15 | Acetylfuran | 95, 110 |
| 16 | Cyclopentandione | 69, 98 |
| 17 | Cyclohexenone | 68, 96 |
| 18 | Propylbenzene | 120 |
| 19 | 3-Methyl-2-cyclopentenone | 96 |
| 20 | 3-Methylbenzene + Unknown (43/85) | 120 |

**Table 1. Compounds identified in the pyrolysate[a]**

| | | |
|---|---|---|
| 21 | Phenol | 94 |
| 22 | $C_{3:1}$ Benzene + 3-Methylbenzene | 118, 120 |
| 23 | Cyclohexandione (70,112) | 70, 112 |
| 24 | Octanal | 128 |
| 25 | Methylcyclopentandione (69,112) | 112 |
| 26 | Limonene (?) | 136 |
| 27 | Indane | 118 |
| 28 | 2,3-Methylcyclopentenone (67,110) | 110 |
| 29 | Indene | 116 |
| 30 | Methyl-propylbenzene | 134 |
| 31 | Methylphenol | 108 |
| 32 | $C_4$ Benzene | 134 |
| 33 | $C_3$ Cyclopentenone ? (109,124) | 124 |
| 34 | Acetophenone | 120 |
| 35 | Methylbenzaldehyde | 120 |
| 36 | Methylbenzaldehyde | 120 |
| 37 | $C_{3:1}$ Benzene + $C_3$ Benzene | 132 + 134 |
| 38 | 4-Ethenyl, $C_2$-benzene | 132 |
| 39 | Nonenol (no 114) | 142 |
| 40 | C2 Phenol | 122 |
| 41 | 1-Acetylcyclohexene + $C_1$-Benzofuran | 124 + 132 |
| 42 | 3-ethyl-2-hydroxy-2-cyclopentenone | 126 |
| 43 | Methylindene | 130 |
| 44 | $C_5$ Benzene | 146 |
| 45 | Naphthalene | 128 |

**Table 1. Compounds identified in the pyrolysate[a]**

| 46 | Trimethylphenol (121, 136) | 136 |
|----|----------------------------|-----|
| 47 | $C_2$ Benzofuran | 146 |
| 48 | $C_2$-Indene | 144 |
| 48 | Indanone | 132 |
| 49 | Methylnaphthalene | 142 |
| 50 | Naphthalenone | 146 |
| 51 | Me-Indanone | 146 |
| 52 | $C_2$-Indanone | 160 |
| 53 | 1,3 Diphenylpropane | 196 |

[a]Peak numbers refer to Fig. 5


00

05

**Table 2. Compounds identified in the thermochemolysate[a]**

| # | Name | M+ |
|---|------|-----|
| 1 | Trimethylamine - TMAH product | 59 |
| 2 | Cyclopentadiene | 66 |
| 3 | $C_{3:1}$ Fatty acid - Me | 86 |
| 4 | Cyclohexadiene | 80 |
| 5 | Benzene | 78 |
| 6 | Methyl-cyclohexadiene | 94 |
| 7 | Dimethylamino-acetonitrile - TMAH product | 84 |
| 8 | Toluene | 92 |
| 9 | Ethylbenzene | 106 |
| 10 | 1,3 + 1,4 Dimethylbenzene | 106 |
| 11 | Styrene + 1,2 Dimethylbenzene | 104 + 106 |
| 12 | 2-Methyl-2-cyclopenten-1-one | 96 |
| 13 | Isopropylbenzene | 120 |
| 14 | Propenylbenzene | 120 |
| 15 | Propylbenzene | 120 |
| 16 | 1-Methyl-3-ethylbenzene | 120 |
| 17 | 1-Methy-4-ethylbenzene | 120 |
| 18 | 1,3,5-Trimethylbenzene | 120 |
| 19 | Methylstyrene + 1-Methyl-2-ethylbenzene | 118 + 120 |
| 20 | Methylstyrene | 118 |
| 21 | Indane | 118 |

**Table 2. Compounds identified in the thermochemolysate[a]**

| 22 | Methoxyphenol | 108 |
|----|---------------|-----|
| 23 | Indene | 116 |
| 24 | $C_4$ Benzene | 134 |
| 25 | Acetophenone | 120 |
| 26 | $C_4$ Benzene | 134 |
| 27 | $C_{4:1}$ Benzene | 132 |
| 28 | Dimethoxybenzene (123, 138) | 138 |
| 29 | $C_1$ Indene | 130 |
| 30 | Naphthalene | 128 |
| 31 | Methylbenzoicacid | 150 |
| 32 | $C_2$ Indene | 144 |
| 33 | Indanone | 132 |
| 34 | Methylnaphthalene | 142 |
| 35 | Methoxybenzoicacid - Me | 166 |
| 36 | Trimethoxybenzene | 168 |
| 37 | $C_3$ Indene | 158 |
| 38 | Diaromatic $C_{12}H_{10}$ | 154 |
| 39 | Ethylnaphthalene | 156 |
| 40 | $C_2$ Naphthalene | 156 |
| 41 | $C_{2:1}$ Naphthalene | 154 |
| 42 | Acenaphthylene | 152 |
| 43 | Dibenzoicacid-Me | 194 |
| 44 | $C_3$ Naphthalene | 170 |

**Table 2. Compounds identified in the thermochemolysate[a]**

| | | |
|---|---|---|
| 45 | Methyl diaromatic | 168 |
| 46 | Dimethoxybenzoicacid-Me | 196 |
| 47 | Diaromatic $C_{13}H_{10}$ (e.g. Fluorene) | 166 |
| 48 | Triaromatic $C_{14}H_{10}$ | 178 |

[a]Peak numbers refer to Fig. 8.

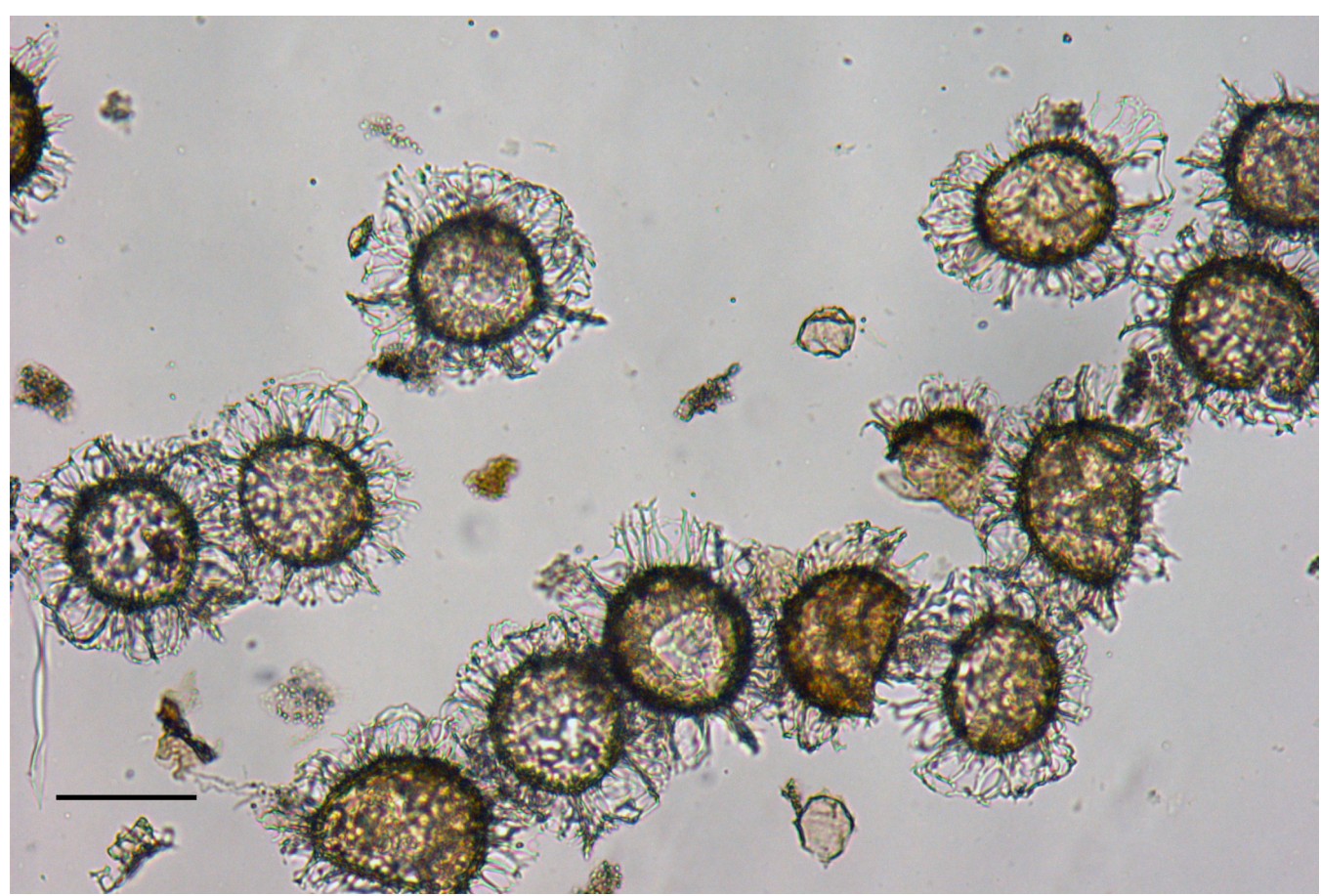

Figure 1: **Microphotograph of the *Thalassiphora pelagica* assemblage from the Kerguelen Plateau prior to purification with a sieve with 50 μm pore size. Scale bar = 100 μm.**

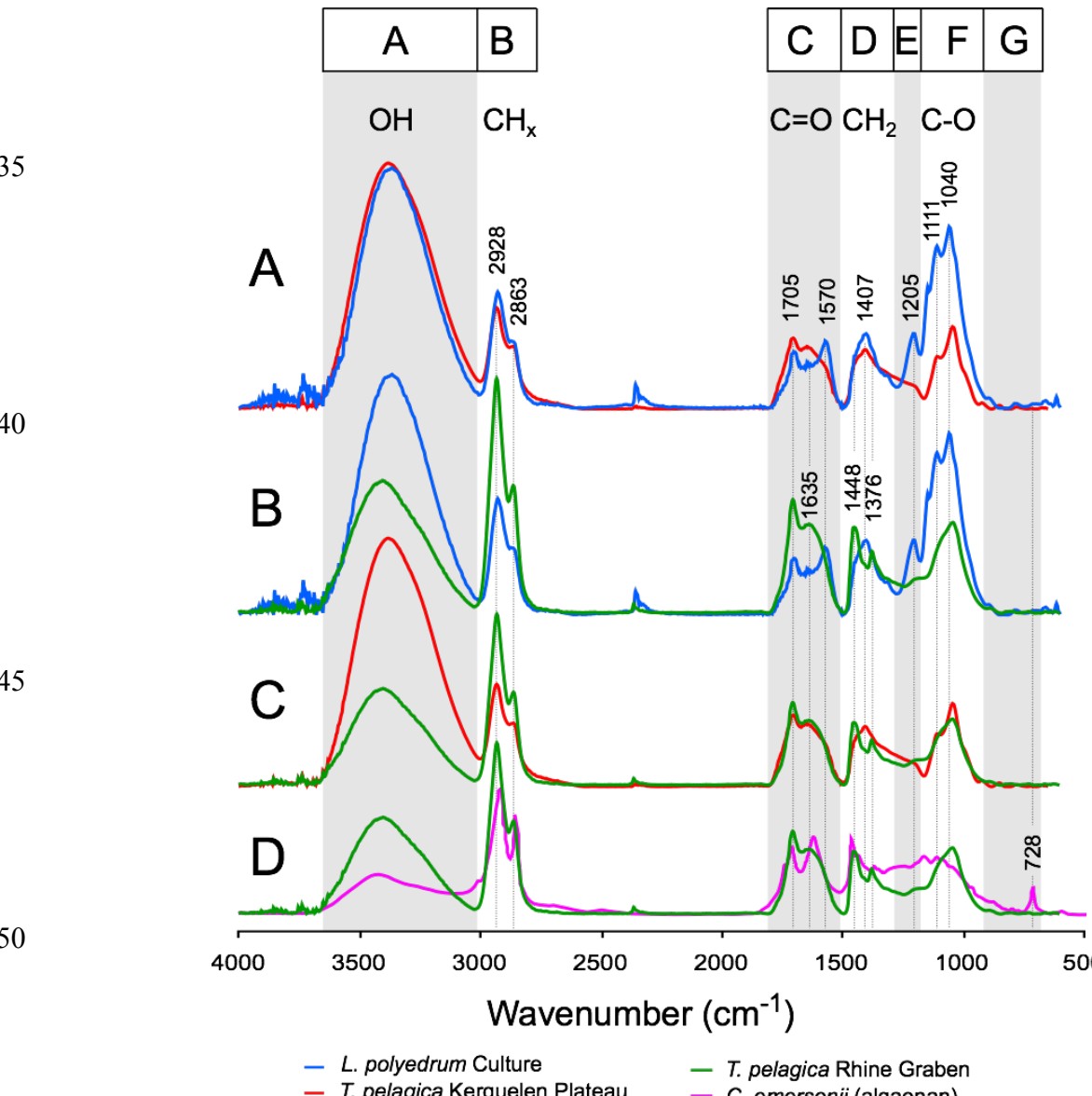

Figure 2: **Pairs of micro-Fourier transform infra red ATR spectra of algae.** *Lingulodinium polyedrum* **cysts from culture (blue line), re-measured sample of Versteegh et al. (2007);** *Thalassiphora pelagica* **cysts from the Kerguelen Plateau (red line);** *Thalassiphora pelagica* **from the Rhine Graben (green line) remeasured sample of Versteegh et al., (2007);** *Clorella emersonii* **(orange line) (adapted from Allard, and Templier, 2000). For further explanation see text.**

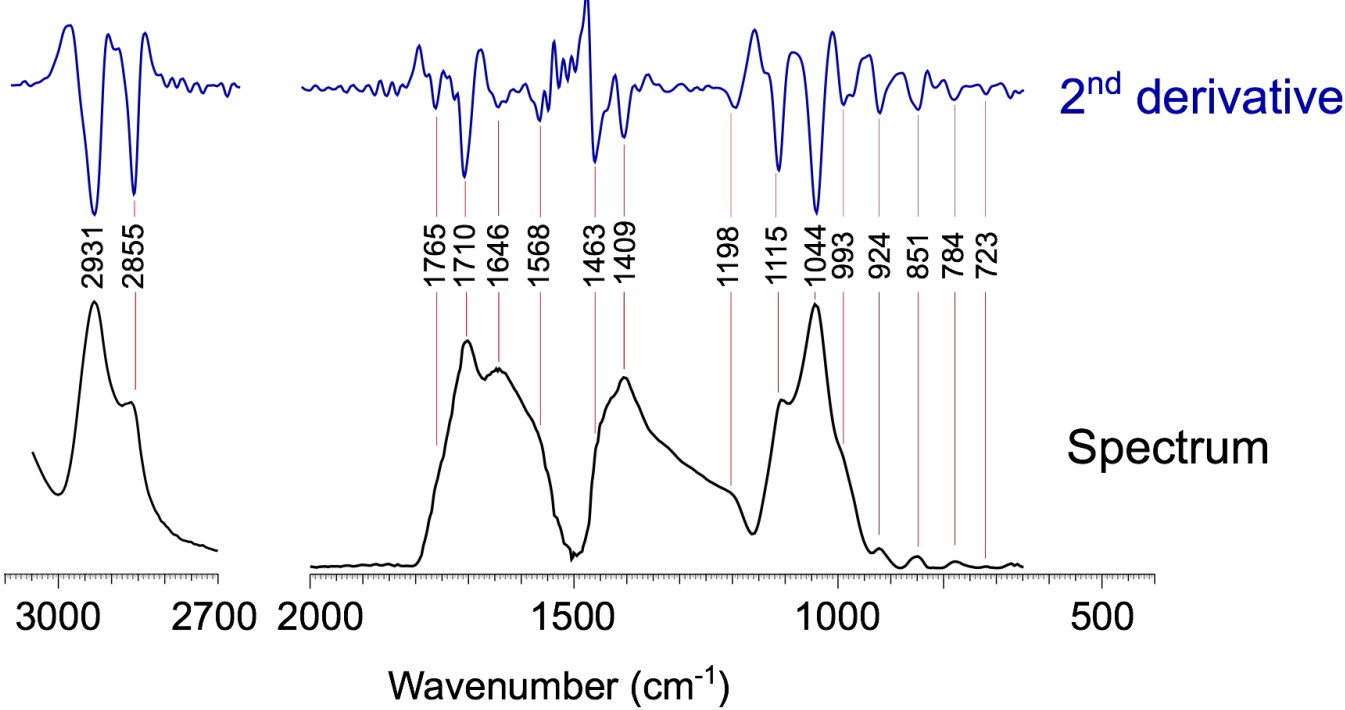

Figure 3: **Detailed views of the fourier transform infra red spectrum of *Thalassiphora pelagica* from the Kerguelen Plateau with second derivative. Top Panel, detail of the 3500-2500 cm$^{-1}$ region of the FTIR absorption spectrum (transmission mode) of *T. pelagica*. Lower panel, detail of the 2000—700 cm$^{-1}$ region with the maxima of the spectrum 2nd derivative showing the locations of absorption bands.**

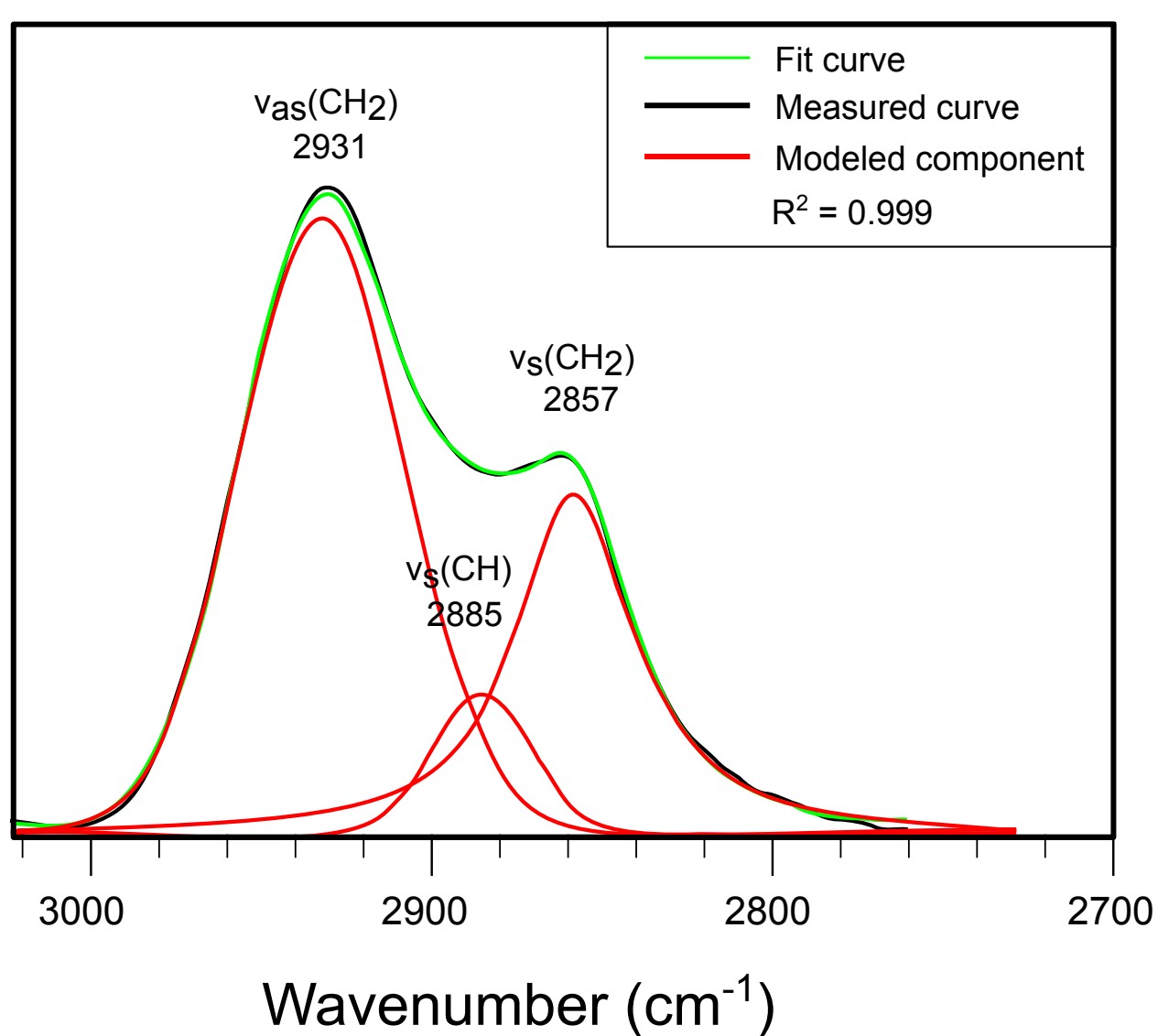

80    Figure 4: **Deconvolution of the CH$_x$ absorption bands for the 3100-2700 cm$^{-1}$ region demonstrating the absence of significant CH$_3$ absorption. Modelled curves are Lorenzian (curve at 2931 cm$^{-1}$) and Gaussian (the other curves).**

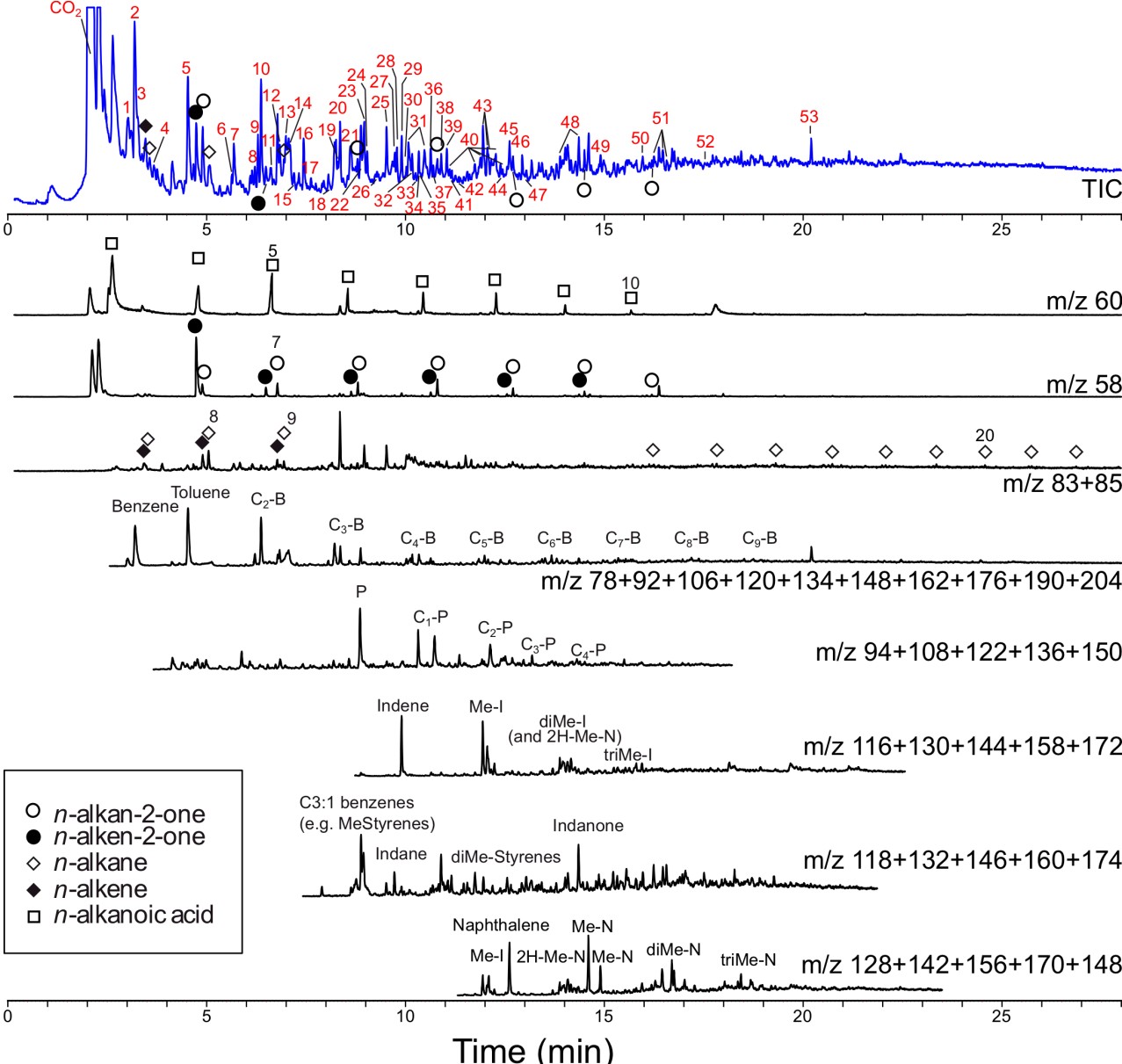

Figure 5: **Chromatogram and mass chromatograms of pyrolysed *Thalassiphora pelagica* cysts from the Kerguelen Plateau. For the total ion current (TIC) the compound names corresponding to numbered peaks (red numbers) are given in Table 1. For the mass chromatograms B=benzene, P=Phenol, N= Naphthalene, I= Indene, 'C-with-number' indicates the number of carbon atoms added to the basic structure.**

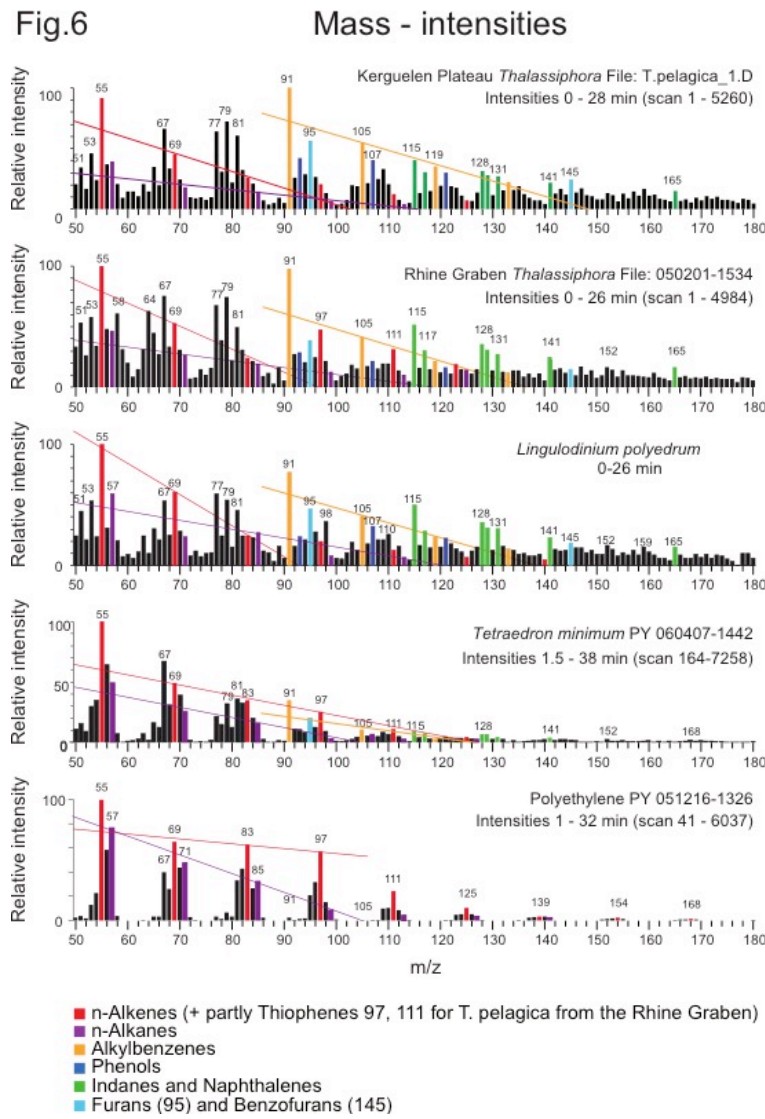

**Fig.6** Mass - intensities

n-Alkenes (+ partly Thiophenes 97, 111 for T. pelagica from the Rhine Graben)
n-Alkanes
Alkylbenzenes
Phenols
Indanes and Naphthalenes
Furans (95) and Benzofurans (145)

95

Figure 6: **Mass intensity plots of the total chromatograms of (a)** *Thalassiphora pelagica* **from the Kerguelen Plateau, (b)** *Thalassiphora pelagica* **from the Rhine Graben (Versteegh et al., 2007), (c) Recent** *Lingulodinium polyedrum* **(Versteegh et al., 2012), (d)** *Tetraedron minimum* **and (e) polyethene. The lines connect masses indicative for** *n*-alkenes (*m/z* 69, 83, 97), *n*-alkanes ( *m/z* 57, 71, 85, 99) **and alkylbenzenes (***m/z* 105, 119) **, respectively. Note that the total ion current of polyethene is dominated by**

00 **series of ions meeting the criteria $C_nH_{2n-1}$ and $C_nH_{2n+1}$, reflecting the highly aliphatic nature of the macromolecule. Note for** *T. pelagica* **the relatively high aromatic contribution (orange bars; m/z 91, 105, 115, 117, 128, 129, 131) in comparison to the 'aliphatic-derived' fragments of alkanes and alkenes, and for** *T. pelagica* **from the Rhine Graben the high relative contribution of m/z 97 and 111 as compared to m/z 69 and 83 which is attributed to the presence of alkylthiophenes.**

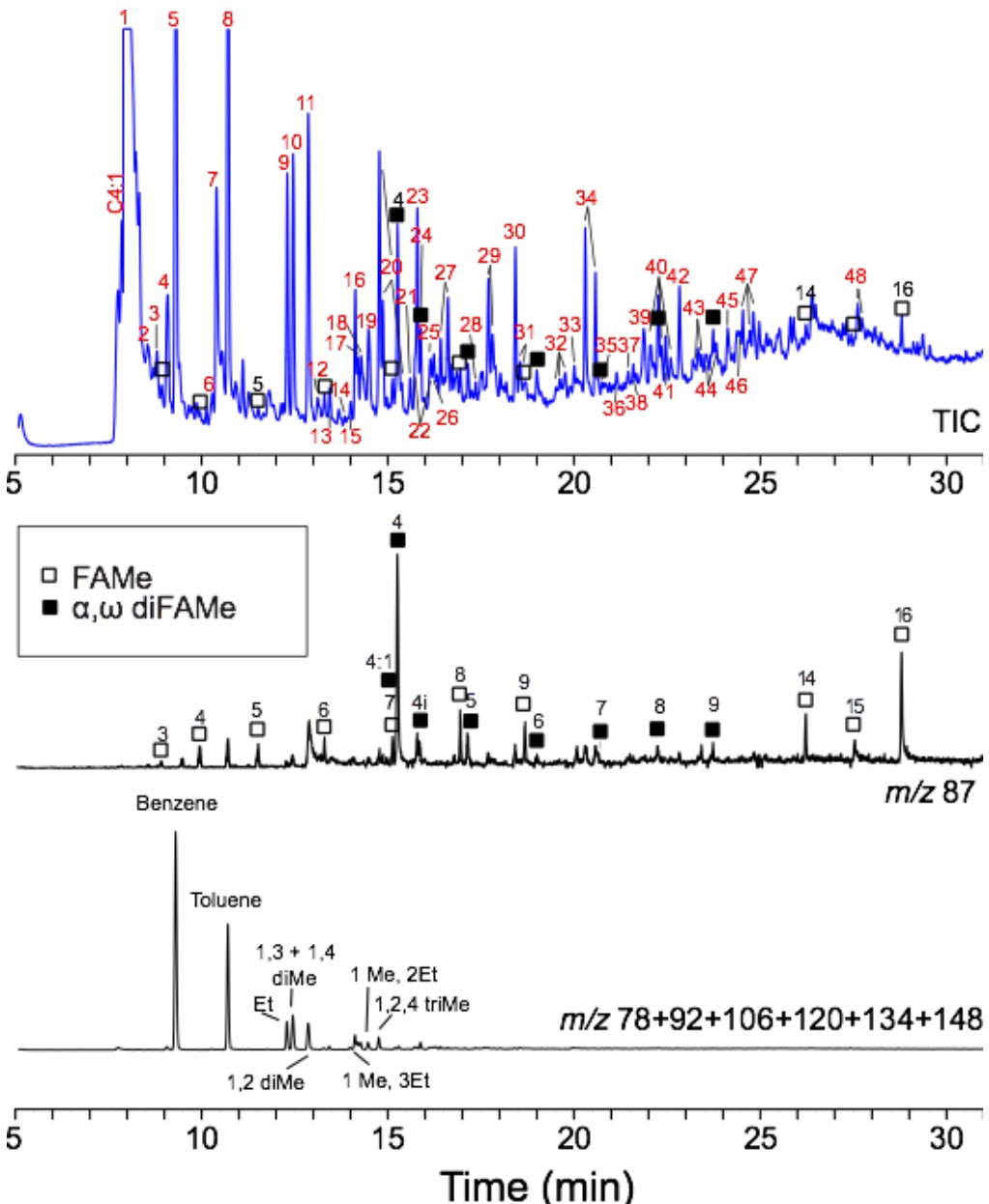

Figure 7: **Chromatogram and mass chromatograms of *Thalassiphora pelagica* cysts from the Kerguelen Plateau after THM-GC-MS. Open squares *n*-alkanoic acids; closed squares *n*-alkanedioic acids. Black numbers indicate the number of carbon atoms of the compound. Red numbers refer to compound names listed in Table 2. Upper panel total ion current (TIC).; Lower panel mass chromatogram of m/z = 87 showing methyl esters of *n*-alkanoic acids and *n*-alkanedioic acids; 4:1 refers to the butenedioic acid, 4i refers to the iso-butanedioic acid. FAMe stands for fatty acid methyl ester.**

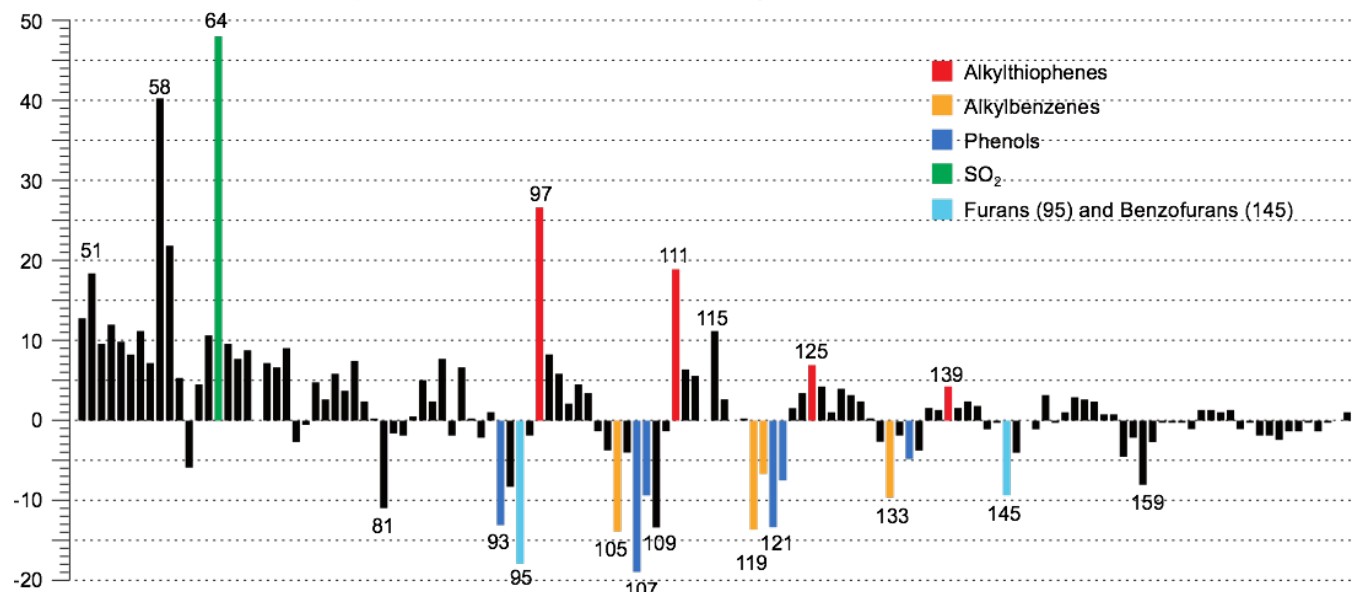

m/z intensities: *T. pelagica* Rhine Graben minus Kerguelen m/z 50-180, 0-28 min.

NB: possible *m/z* 58 fragments: $C_3H_6O$, $C_4H_{10}$, $C_2S$

Figure 8: **Relative mass intensities of the total chromatograms of *Thalassiphora pelagica* from the Rhine Graben minus those of *Thalassiphora pelagica* from the Kerguelen Plateau (Versteegh et al., 2007). Masses with abundances < 0 are more abundant in the specimens from the Kerguelen Plateau. Note the masses indicating sulphur containing fragments (thiophenes) are more important in the specimens from the Rhine Graben whereas masses characterizing oxygen containing fragments (furans, phenols) and alkylbenzenes are more abundant in specimens from the Kerguelen Plateau.**

## Supplementary Information

Published infrared spectra of the cysts of *L. polyedrum* and *T. pelagica* had been obtained by means of different spectrometric methods and using different brands and types of equipment. To obtain a better insight in the extent to which differences between spectra resulted from differences in cyst structure or differences in measurement conditions we traced the original material and re-measured all cysts using the same analytical procedure. Supplementary figures 1 and 2 demonstrate the influence of different
methods of measurement.

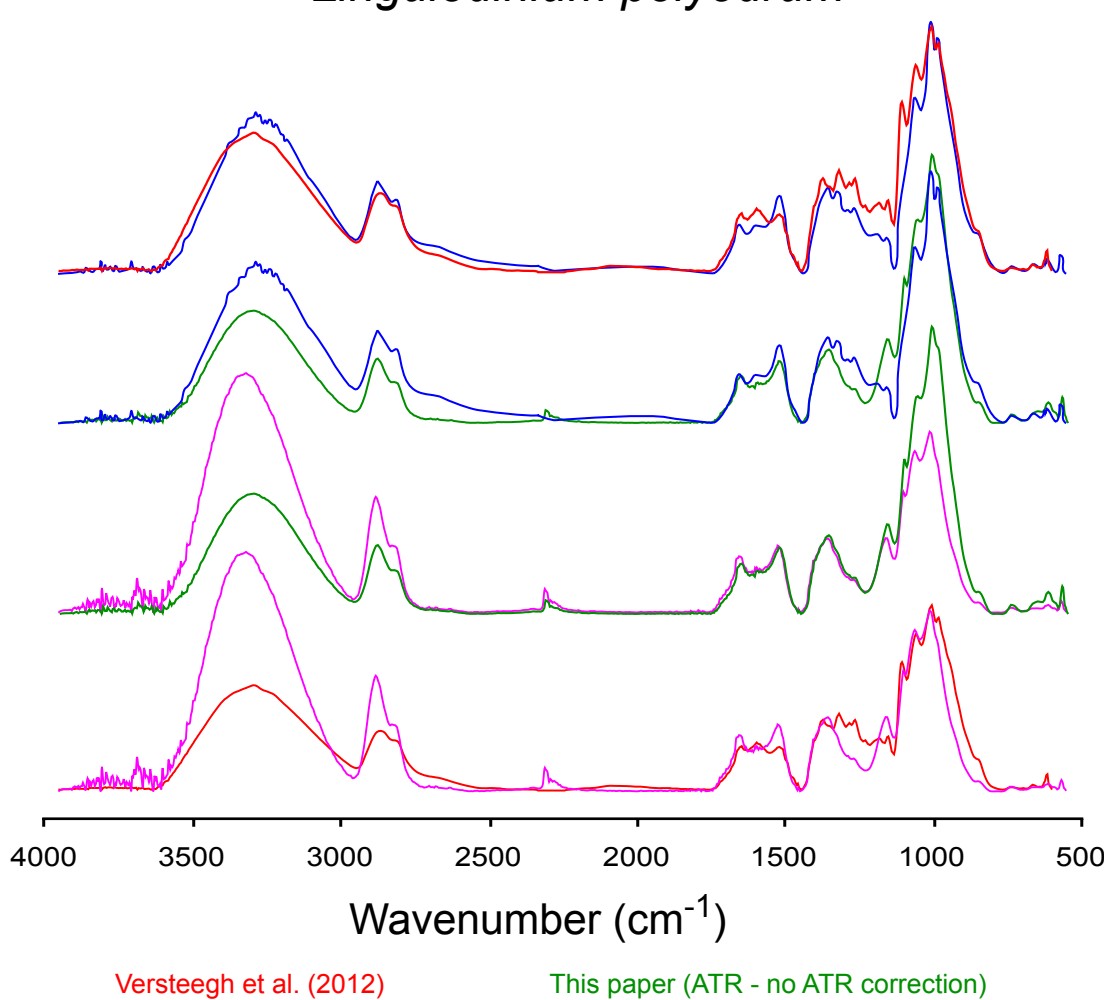

## *Lingulodinium polyedrum*

Wavenumber (cm$^{-1}$)

Versteegh et al. (2012)     This paper (ATR - no ATR correction)
This paper (reflection)     This paper (ATR - ATR correction)

Supplementary Figure 1: **Comparison of published and re-measured infrared spectra derived from culture derived** *Lingulodinium polyedrum* **cysts measured in reflection mode and micro ATR mode. All spectra are derived from same sample, and received the same sample processing.**


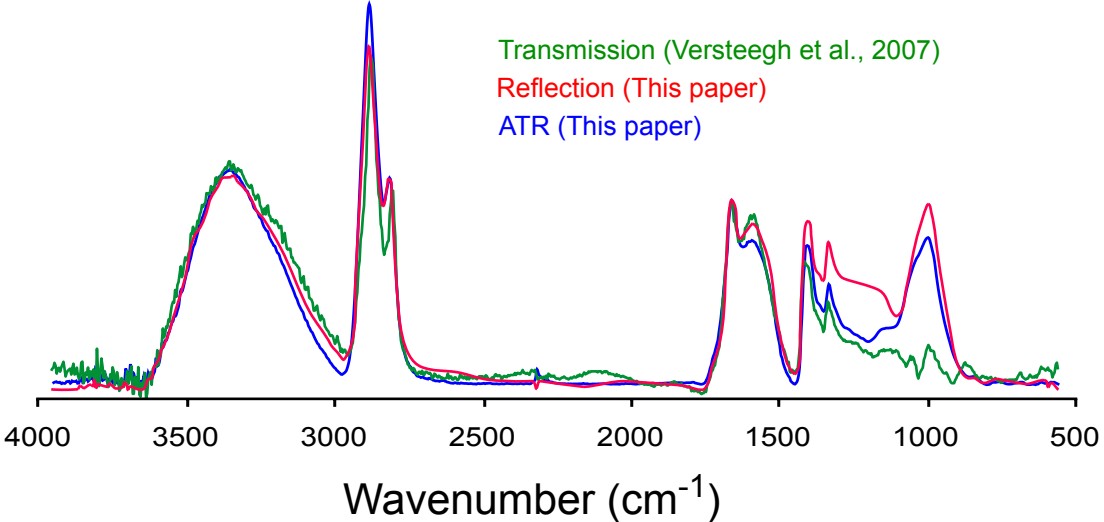

## *Thalassiphora pelagica* - Rhine Graben

Transmission (Versteegh et al., 2007)
Reflection (This paper)
ATR (This paper)

Wavenumber (cm$^{-1}$)

65 Supplementary Figure 2: **Comparison of published and re-measured infrared spectra derived from *Thalassiphora pelagica* cysts from the Rhine Graben, measured in transmission, reflection and micro ATR modes. All spectra are derived from same sample, and received the same sample processing.**