# Peer review of "Better molecular preservation of organic matter in an oxic than in a sulphidic depositional environment: evidence from of *Thalassiphora pelagica* (Dinoflagellata, Eocene) cysts."

_Biogeosciences, 2019_

## Referee Comment (RC1) · Morgan Raven (Referee) · 24 Nov 2019

In this manuscript, the authors present a detailed characterization of dinoflagellate cyst walls that were deposited in oxic versus sulfidic sediments. This data directly inform a critical and timely knowledge gap, as they address the chemical mechanisms driving organic matter preservation as well as taphonomic issues. The dataset will be a significant contribution and I support its eventual publication in BGD. Before that point, however, I encourage the authors to significantly expand their discussion of the broader implications of their results, especially relating to selective biases in the fossil record

and the importance of organic matter sulfurization versus other aspects of preservation under anoxic conditions.

Specific notes:

Broadly speaking, many paragraphs would benefit from the addition of a clear concluding sentence, summarizing the main point or argument arising from the preceding results.

Figure 2: One key thing the reader needs to assess here is whether the green line or the black line is more similar to the blue line, which is difficult to do in this arrangement. Consider rearranging your figure so the blue line is common among the sections. The common line also appears stretched at different scales, making it particularly difficult to cross compare the green vs blue lines.

Line 49 – Please revise this sentence: "in the absence of reactive Fe, which has the potential to outcompete organic molecules for reactive polysulfides and limit organic matter sulfurization".

Line 51 – , which

Line 69 –To broader the accessibility of your results, provide a description of the key aspects of pelagica cyst composition here. Why is this organism relatively resistant to aerobic degradation, and what does that mean for the interpretation of your results?

Line 74 – I'm not sure what you mean by "the addition of carboxylic acids by early sulfurization of the cysts." To what molecule is the reduced sulfur being added? What is the source of the carboxylic acids?

Line 83 – For both sites, please provide a description of the environment, including a summary of what is known about sedimentation rates. How do the burial ages of these samples differ? What other sedimentological differences exist between the sites? (e.g., clays vs biogenic silica, overall TOC, water depth etc). This information is critical to holistically assess the possible drivers of variation between the two sites. I would hope

to see this section substantially expanded.

Line 85 –Please provide a complete description of the processing of the reducing cysts here. The 2007 paper is behind a paywall and not available to all of your readers.

Line 110 – Either write out the description of your conditions in sentences or use a table.

Line 112 – What internal standards were used to verify and track retention times? Were any of the identified compounds in your table 1 confirmed with authentic standards? Inclusion of such standards would significantly enhance confidence in the contents of Tables 1+2.

Line 114 – The paywall, again: please make your methods self-contained to this manuscript.

Line 122 – I would find this information (band definitions, minima, etc) more useful in table format. A lot of your section 4.1.1 reads like results rather than discussion; consider moving that results interpretation up to this point. This will also allow your discussion section to focus on the exciting takeaways rather than more routine IDs.

Line 159 – marked absence of absorptions by CH3 – this seems important and is highlighted in Figure 4, but the significance of this observation is not really discussed. What does this mean?

Section 4.1.1 – Reads like results.

Line 204-207 – There are multiple sentences starting with "this" or "it" in this section, which makes it difficult to precisely follow the argument. Please revise for clarity.

Line 205 – please revise sentence "and which for its absence in recent cysts we attribute"

Line 207 – What else is known about this rapid C=O addition phenomenon? What is the proposed mechanism? (It is an oxic phenomenon?)

Line 213 – I like the way this is set up, and I agree chemical processes are likely dominant. But, there are also geological / sedimentological processes that may differ between the sites, most notably sedimentation rate and the composition of surrounding sediments. Please address these potential differences and explain why they do not explain the contrasts in cyst composition you observe.

Line 218 – sentence: "Specifically, the Rhine Graben pelagica lack the absoption feature at 718cm-1 and thus appear to lack long chains of algaenan, contrasting pelagica from the Kerguelan Plateau."

Section 4.1.3 – Please add a concluding sentence to this section that summarizes your case for how FTIR spectra show that redox differences in the sedimentary environment are the key driver of differences in Figure 2.

"A further alteration in the same direction as the differences" is unclear. Please be as specific as possible about what observations you're talking about.

Line 220 – sentence more like: "GC-MS results are consistent with the results of FTIR analyses, which suggest that differences in depositional environment are associated with differences in molecular structure"

Line 225-227 – Please revise this sentence to clearly explain which group of alkanes were bound by which mechanism and where those pools ended up in your analytical flow.

Line 235 – define "it" (in "its presence") their?

Line 239 – Consider driving home this point "sulfurization may thus eliminate/consume carbohydrate hydroxy groups from the cyst walls, leaving __". Indications for the molecular selectivity of sulfurization reactions seems like a significant aspect of your results - please discuss further.

When you find that sulfurized materials are richer in long-chain aliphatics, what does that mean for the relative importance of carbohydrate vs. lipid sulfurization?

Line 240 – define which sample you are discussing in this paragraph.

Line 249 – don't stop! ïĄŁ You've set up some really exciting observations (e.g., Fig. 8) and this feels as though their implications haven't been fully fleshed out yet. Please also clarify your final sentence – how exactly do you see aliphatic content fitting into the sulfurization story?

Can you say anything further about the signature of sulfurization in the geologic record? How cyst sulfurization would bias the interpretation of fossils? Which general categories of molecules would be most susceptible to alteration (e.g., carbohydrates, maybe lipids less so)? Many of these ideas seem to be hidden within the text but would benefit from being explicitly stated.

Finally, I would strongly advocate for the authors to include at least a rough quantification of the sulfur content of these samples, for example by combustion elemental analyzer or x-rays. Sulfur-to-carbon molar ratios would be extremely valuable for the identification of similar processes in other environments. Sulfur quantification would also allow you to assess to what extent sulfurization can explain the alteration of your bulk organic matter or whether some other aspect of a reducing environment might have been the main driver (for example, you could compare whether the sulfur addition is sufficient to account for the loss of hydroxyl functional groups that you observe.)

---

## Referee Comment (RC2) · Jakob Vinther (Referee) · 15 Dec 2019

The authors identify that samples in euxinic conditions have undergone substantial kerogenisation, while samples in 'oxic' settings have not.

Their argument that the oxic conditions are better seems apparently good on these premises. I would like to stress, though, that I am not certain about the premise of good preservation and also what constitute oxic environments.

I have two key issues:

[Figure]

1. Kerogenisation is not necessarily bad organic preservation. This may be a way to quench many labile organic molecules into larger macromolecules. What they mean is that the cysts have been overprinted by kerogenisation. This is not bad organic preservation per se, but not good if you want to look at the original composition of a microfossil.

2. When the authors call the other deposit oxic I am not sure that this has been proven and in fact, I doubt it is fully oxic throughout. While it may be on the surface, almost all sediments switch to anoxic conditions quickly in the subsurface. Dinoflagellates are robust and survive initial oxic decay under almost any circumstances contrary to most other tissues. This is the reason why exceptional Konservat Lagrstättens are notoriously anoxic environments. But, this does not mean that oxic environments may not preserve extremely stable and recalcitrant biomolecules such as dinoflagellates and pollen as it will switch to anoxic conditions a few centimeters below the sediment-water interface.

In conclusion. I would like the authors to nuance in their abstract, title and throughout the distinction between kerogenisation and in situ polymerisation with respect to preservation without it. Finally, I don't think that the dichotomy between 'euxinic' and 'oxic' is true given that sediments quickly become anoxic soon after deposition and dinoflagellates would survive the initial oxic conditions that would have been existing in the subsurface.

Therefore, I would focus on the nature of kerogenisation and how this complicates investigation of original biosignatures endogenous to a microfossil. Describe and discuss kerogenisation and therefore how an investigation of tissues in euxinic settings need to evaluate degrees of kerogenisation before doing other chemical analyses, such as isotope composition or more superficial chemical analyses, such as FTIR, RAMAN, TOF SIMS or similar, as they would characterise a mixed bag of molecules. This, I think, would make for an interesting comparison and a study I would find useful.

Best wishes Jakob

---

## Author Comment (AC1) · 12 Mar 2020

by **Morgan Raven (Referee)**

In this manuscript, the authors present a detailed characterization of dinoflagellate cyst walls that were deposited in oxic versus sulfidic sediments. This data directly inform a critical and timely knowledge gap, as they address the chemical mechanisms driving organic matter preservation as well as taphonomic issues. The dataset will be a significant contribution and I support its eventual publication in BGD. Before that point, however, I encourage the authors to significantly expand their discussion of the broader implications of their results, especially relating to selective biases in the fossil record and the importance of organic matter sulfurization versus other aspects of preservation under anoxic conditions.

We considerably expanded the manuscript taking these suggestions into account. However, we like to stress that the paper is not about the selective bias due to sulfurization alone. It rather highlights the differences in modification of refractory organic matter (dinosporin) between an aerobic and an anaerobic setting. To do this we present new information on the aerobic setting and add this to the published information already available for the anaerobic setting.

Specific notes:

Broadly speaking, many paragraphs would benefit from the addition of a clear concluding sentence, summarizing the main point or argument arising from the preceding results. Such sentences have been added.

Figure 2: One key thing the reader needs to assess here is whether the green line or the black line is more similar to the blue line, which is difficult to do in this arrangement. Consider rearranging your figure so the blue line is common among the sections. The common line also appears stretched at different scales, making it particularly difficult to cross compare the green vs blue lines. We revised this figure completely an also remeasured spectra.

Line 49 – Please revise this sentence: "in the absence of reactive Fe, which has the potential to outcompete organic molecules for reactive polysulfides and limit organic matter sulfurization". Done

Line 51 – , which Done

Line 69 –To broader the accessibility of your results, provide a description of the key aspects of pelagica cyst composition here. Why is this organism relatively resistant to aerobic degradation, and what does that mean for the interpretation of your results? We added a few lines on the aerobic and anaerobic degradation of dinoflagellate cysts. Since cysts sensitive to aerobic degradation degrade quickly (Zonneveld et al., 2019 and references therein) only resistant cysts remain in aerobic settings. The resistance means that we can do this comparison. In any other case, the cysts would be gone from the Kerguelen sample.

Line 74 – I'm not sure what you mean by "the addition of carboxylic acids by early sulfurization of the cysts." To what molecule is the reduced sulfur being added? What is the source of the carboxylic acids? Has been explained now.

Line 83 – For both sites, please provide a description of the environment, including a summary of what is known about sedimentation rates. How do the burial ages of these samples differ? What other sedimentological differences exist between the sites? (e.g., clays vs biogenic silica, overall TOC, water depth etc). This information is critical to holistically assess the possible drivers of variation between the two sites. I would hope to see this section substantially expanded. This section has been expanded now and a description of age, sedimentation rate, redox environment, TOC content and general environmental setting is now present.

Line 85 –Please provide a complete description of the processing of the reducing cysts here. The 2007 paper is behind a paywall and not available to all of your readers. Info has been added.

Line 110 – Either write out the description of your conditions in sentences or use a table. Done

Line 112 – What internal standards were used to verify and track retention times? Were any of the identified compounds in your table 1 confirmed with authentic standards? Inclusion of such standards would significantly enhance confidence in the contents of Tables 1+2. A mixture of n-alkane standards was used to calibrate the retention times. Over the years we produced a considerable library of compounds, spectra and relative retention times. During this period again and again, pyrolysis products were checked with internal standards (such as n-alkanes, n-alcohols, n-carboxylic acids, a wide range of linear and cyclic isoprenoids, alkylated aromatics, carbohydrates, polycyclic aromatics. Identification is based this accumulated knowledge. We made this more clear in the text

Line 114 – The paywall, again: please make your methods self-contained to this manuscript. Information has been added

Line 122 – I would find this information (band definitions, minima, etc) more useful in table format. A lot of your section 4.1.1 reads like results rather than discussion; consider moving that results interpretation up to this point. This will also allow your discussion section to focus on the exciting takeaways rather than more routine IDs. We moved section 4.1.1. into the results section. As a result the former result section largely became obsolete and has been removed. We therefore also refrain from adding a table.

Line 159 – marked absence of absorptions by CH3 – this seems important and is highlighted in Figure 4, but the significance of this observation is not really discussed. What does this mean? We elaborated more on this issue.

Section 4.1.1 – Reads like results. see reply to remark for line 122

Line 204-207 – There are multiple sentences starting with "this" or "it" in this section, which makes it difficult to precisely follow the argument. Please revise for clarity. Done.

Line 205 – please revise sentence "and which for its absence in recent cysts we at- tribute" Done

Line 207 – What else is known about this rapid C=O addition phenomenon? What is the proposed mechanism? (It is an oxic phenomenon?)We rephrased the sentence so that it is clear now this is an oxic mechanism.

Line 213 – I like the way this is set up, and I agree chemical processes are likely dominant. But, there are also geological / sedimentological processes that may differ between the sites, most notably sedimentation rate and the composition of surrounding sediments. Please address these potential differences and explain why they do not explain the contrasts in cyst composition you observe. A good point. We now provide a more thorough explanation of the geological settings and their implications.

Line 218 – sentence: "Specifically, the Rhine Graben pelagica lack the absoption feature at 718cm-1 and thus appear to lack long chains of algaenan, contrasting pelagica from the Kerguelan Plateau."  Could not find this in the pdf

Section 4.1.3 – Please add a concluding sentence to this section that summarizes your case for how FTIR spectra show that redox differences in the sedimentary environment are the key driver of differences in Figure 2.  We added a small concluding paragraph to what is now 4.2

Line 213 "A further alteration in the same direction as the differences" is unclear. Please be as specific as possible about what observations you're talking about. The section has been rewritten

Line 220 – sentence more like: "GC-MS results are consistent with the results of FTIR analyses, which suggest that differences in depositional environment are associated with differences in molecular structure" OK, changed

Line 225-227 – Please revise this sentence to clearly explain which group of alkanes were bound by which mechanism and where those pools ended up in your analytical flow.  Done

Line 235 – define "it" (in "its presence") their? Done

Line 239 – Consider driving home this point "sulfurization may thus eliminate/consume carbohydrate hydroxy groups from the cyst walls, leaving __". Indications for the molecular selectivity of sulfurization reactions seems like a significant aspect of your results - please discuss further. In this case the text is about the oxidized cysts from the Kerguelen Plateau. Here the diagenetic processes presumably removed the hydroxyl groups of the carbohydrate, so that their methylation upon thermochemolysis with TMAH did not occur to such an extent.

When you find that sulfurized materials are richer in long-chain aliphatics, what does that mean for the relative importance of carbohydrate vs. lipid sulfurization? We do not think that we can draw a conclusion on this right away. The depositional environment of the Rhine Graben was such that large amounts of lipids must have reached the sea floor and were stored in the sediment. this kind of environment typically develops into a source rock for oil and gas. For the kerguelen plateau, these lipids have probably already been degraded during their transport through the oxic water column to the sediment floor.

Line 240 – define which sample you are discussing in this paragraph. Done

Line 249 – don't stop! You've set up some really exciting observations (e.g., Fig. 8) and this feels as though their implications haven't been fully fleshed out yet. Please also clarify your final sentence – how exactly do you see aliphatic content fitting into the sulfurization story? We added a paragraph just before the conclusions clarifying this in more detail.

Can you say anything further about the signature of sulfurization in the geologic record? How cyst sulfurization would bias the interpretation of fossils? Which general categories of molecules would be most susceptible to alteration (e.g., carbohydrates, maybe lipids less so)? Many of these ideas seem to be hidden within the text but would benefit from being explicitly stated.

We now discuss these questions in the manuscript.

The cyst sulphurisation is only apparent on molecular level, not visible at microscopic level and as such does not provide bias. Furthermore, the cysts belong intrinsically to the more refractory organic matter and the sulphurisation does not increase their preservation and therefore presence in the fossil record.

Our study does not provide an answer to the type of molecules that is most vulnerable to sulphurisation, it rather describes how the dinoflagellate cyst wall gets modified differently in different environments. The kind of interactions in both environments that are apparent are polymerization reactions with linear aliphatics, simply since these are not available in the initial biomacromolecule. We did implement answers to these questions in the revised manuscript.

Finally, I would strongly advocate for the authors to include at least a rough quantification of the sulfur content of these samples, for example by combustion elemental analyzer or x-rays. Sulfur-to-carbon molar ratios would be extremely valuable for the identification of similar processes in other environments. Sulfur quantification would also allow you to assess to what extent sulfurization can explain the alteration of your bulk organic matter or whether some other aspect of a reducing environment might have been the main driver (for example, you could compare whether the sulfur addition is sufficient to account for the loss of hydroxyl functional groups that you observe.)

We agree that it would be interesting to do these analyses. However, how interesting the questions that may be attacked we consider our non-quantitative study not suited to efficiently attack these questions and therefore the proposed analyses beyond the scope of this paper.

---

## Author Comment (AC2) · 12 Mar 2020

Dear Editor,

From the constructive comments by Jacob Vinther, we understand that we were not clear in what we think happened to the microfossils. We implemented the suggestion to better separate between intermolecular and intramolecular interactions. In our case, the former primarily adding molecules to the initial cyst biomacromolecule, the latter modifying the initial structure. Their added effects determine the modification from cyst

bio- to geomolecule. To address these points we adapted the text to better elucidate these issues. We disagree to some extent with the suggested terminology in our case - kerogenisation and in-situ polymerization. With respect to the oxygen exposure of the OM for the Kerguelen Plateau sample we added information on the depositional environment. Furthermore, we made more clearly where the differences lay between the cysts from the three samples: the modern analogue (published 2012), the Rhine graben (published 2007) and the Kerguelen Plateau (the new information). Our infrared, pyrolysis and thermochemolysis analyses are unanimous and support each other in that the molecular composition of the material from the Kerguelen Plateau differs less from the modern analogue than the material from the Rhine Graben - e.g. the former generates a carbohydrate signature upon thermochemolysis, the latter does not). On the basis of this evidence we conclude that the molecular preservation of the material preserved in the Kerguelen Plateau is better. Finally there seem to be different concepts (or possibly a semantic issue) with respect the use of 'kerogenisation' and polymerisation when applied to modification of biomacromolecules in a diagenetic context. To avoid these problems we rather refrain from using these as much as possible.

In the attached pdf you will find a detailed account of the changes made.

sincerely,

Gerard Versteegh

Please also note the supplement to this comment:
https://www.biogeosciences-discuss.net/bg-2019-373/bg-2019-373-AC2-supplement.pdf

――――――――――――――――――――

**Supplement:**

by **Jakob Vinther (Referee)**

The authors identify that samples in euxinic conditions have undergone substantial kerogenisation, while samples in 'oxic' settings have not.

Their argument that the oxic conditions are better seems apparently good on these premises. I would like to stress, though, that I am not certain about the premise of good preservation and also what constitute oxic environments.

I have two key issues:

1. Kerogenisation is not necessarily bad organic preservation. This may be a way to quench many labile organic molecules into larger macromolecules. What they mean is that the cysts have been overprinted by kerogenisation. This is not bad organic preservation per se, but not good if you want to look at the original composition of a microfossil.

We adapted the text to make a better distinction between intermolecular and intramolecular diagenetic processes accordingly to make clear what we do mean. We added a paragraph outlining that we consider better and worse molecular preservation, what we consider kerogenisation and polymerization as is explained also below:

As we mention in the text, kerogen has been defined as the sedimentary organic matter that is insoluble in common organic solvents (see Durand, B., 1980. Sedimentary organic matter and kerogen. Definition and quantitative importance of kerogen. In: Durand B., Kerogen : Insoluble Organic Matter from Sedimentary Rocks, Editions Technip, Paris13-34). As such an organic microfossil fits in this definition of kerogen.

Kerogenisation is simply the process of diagenetic cross-linkage (see Butterfield, N. J., 1990. Organic preservation of non-mineralizing organisms and the taphonomy of the Burgess Shale. Paleobiology 16, 272-286). This cross linking may thus be intermolecular and intramolecular.

This definition of kerogenisation is too narrow for what we describe in the manuscript. We consider diagenetic change of the dinoflagellate cyst wall. This involves not only diagenetic cross linking (kerogenisation) but all kinds of structural changes, thus also e.g., modifying, adding and losing bonds and functional groups, changing stereochemistry.

We report on the extent to which the cyst biomacromolecule has been modified by different diagenetic regimes - indeed changes in the original structure of the microfossil biomacromolecule. In that perspective, each change away from the

original biomacromolecule is a decrease in its condition of preservation. We obtain insight in the degree of change through infrared spectroscopy, pyrolysis and thermochemolysis of the cysts and comparison with each other and an unaltered recent analogue. This shows that the cyst preserved in the environment that was subject to extensive aerobic OM mineralization underwent diagenetic changes which were different from the changes to the cysts preserved in the initially anaerobic setting. These changes not only involved diagenetic cross linkage but also addition and removal of functional groups. Unexpectedly, the cysts from the aerobic setting appear structurally more close to the recent analogue than the cysts preserved in the initially anaerobic setting. We do not mention or imply that the preservation is good or bad in and absolute sense, we only mention that the molecular structure of the cysts from the Kerguelen Plateau preserved better at molecular level than those from the Rhine Graben. And we find a reasonable explanation why this is so.

We revised the text such that this becomes more clear.

2. When the authors call the other deposit oxic I am not sure that this has been proven and in fact, I doubt it is fully oxic throughout. While it may be on the surface, almost all sediments switch to anoxic conditions quickly in the subsurface. Dinoflagellates are robust and survive initial oxic decay under almost any circumstances contrary to most other tissues. This is the reason why exceptional Konservat Lagrstättens are notoriously anoxic environments. But, this does not mean that oxic environments may not preserve extremely stable and recalcitrant biomolecules such as dinoflagellates and pollen as it will switch to anoxic conditions a few centimeters below the sediment- water interface.

We agree, we were not clear about this issue. We explain this more clearly now. The depositional setting has been oxic and considered to be an open oceanic setting with low sedimentation rate and carbon lean sediments. Oxygen penetration must have been more than a few centimeters. The oxygen exposure time must have been long and organic matter mineralization extensive. Considering the marine environment, probably the most common state is that the sediments are oxic beyond a few cm from the surface. With oceanic sedimentation rates this accounts easily for centuries to millennia of aerobic degradation which considering the low OM input from above produces OM poor oozes. In large parts of the Pacific oxygen penetration reaches even decimeters to meters below the sediment surface and even may reach the basement. Only in upwelling and shelf regions, mostly near the continental margins, this may be different but also here, bioturbation often significantly extends the oxygen exposure time of the sedimentary organic matter.

In conclusion. I would like the authors to nuance in their abstract, title and throughout the distinction between kerogenisation and in situ polymerisation with respect to preservation without it. Finally, I don't think that the dichotomy between 'euxinic' and 'oxic' is true given that sediments quickly become anoxic soon after deposition and dinoflagellates would survive the initial oxic conditions that would have been existing in the subsurface.

We elucidated that kerogenisation is diagenetic crosslinking as is in situ polymerization. The difference lies lay in the word polymer. A biopolymer is macromolecule that exists of repeating units of relatively simple molecules (biological monomers). In contrast to synthetic polymers small variations between the monomers are allowed (e.g. DNA, polypeptides, polysaccharides). We do not make this distinction for the above-mentioned reasons.

With respect to the preservation of organic-walled dinoflagellates there is a very strong difference in preservation potential between the different groups of dinoflagellates, with the most labile cyst types degrading under suboxic conditions and the most refractory cysts hardly degrading after millennia oxygen exposure (for a recent overview see

Zonneveld, K. A. F. Gray, D. D. Kuhn, G. and Versteegh, G. J. M., 2019. Postdepositional aerobic and anaerobic particulate organic matter degradation succession reflected by dinoflagellate cysts: The Madeira Abyssal Plain revisited. Marine Geology 408, 87-109).

We deliberately analysed a degradation resistant cyst species to avoid the problem that the species would have been entirely mineralized in the Kerguelen Plateau sediments.Therefore, I would focus on the nature of kerogenisation and how this complicates in- vestigation of original biosignatures endogenous to a microfossil. Describe and discuss kerogenisation and therefore how an investigation of tissues in euxinic settings need to evaluate degrees of kerogenisation before doing other chemical analyses, such as iso- tope composition or more superficial chemical analyses, such as FTIR, RAMAN, TOF SIMS or similar, as they would characterise a mixed bag of molecules. This, I think, would make for an interesting comparison and a study I would find useful.

We would like to point out that this paper is not primarily on what happens with organic matter in euxenic settings and that we did FTIR.

Best wishes,

Gerard Versteegh

---

## Author Response (AR2)

Dear Silvio Pantoja,

Thank you for having another close look at the manuscript.

We followed all your editorial suggestions as you will see in the annotated manuscript below.

best wishes,

Gerard Versteegh

[revised manuscript text omitted]

*Lingulodinium polyedrum*

<table>
<tr><td style="color:red">Versteegh et al. (2012)</td><td style="color:green">This paper (ATR - no ATR correction)</td></tr>
<tr><td style="color:blue">This paper (reflection)</td><td style="color:magenta">This paper (ATR - ATR correction)</td></tr>
</table>

70

Supplementary Figure 1: **Comparison of published and re-measured infrared spectra derived from culture derived *Lingulodinium polyedrum* cysts measured in reflection mode and micro ATR mode. All spectra are derived from same sample, and received the same sample processing.**

75

[Figure]

**Thalassiphora pelagica* - Rhine Graben**

Transmission (Versteegh et al., 2007)
Reflection (This paper)
ATR (This paper)

80

Supplementary Figure 2: **Comparison of published and re-measured infrared spectra derived from *Thalassiphora pelagica* cysts from the Rhine Graben, measured in transmission, reflection and micro ATR modes. All spectra are derived from same sample, and received the same sample processing.**

85

The weakly defined absorptions between 1305 and 1203 cm$^{-1}$ (Band E; 1300—1150 cm$^{-1}$) can be related to C-O carboxylic acids.

For band F (1150—900 cm$^{-1}$), the absorptions at 1112, 1047, and 986 cm$^{-1}$ are assigned to C-O e.g. as C-O-C and C-O-H of carbohydrates.

For band G the minor absorption at 728 cm$^{-1}$ is attributable to CH$_2$ rock, its low intensity agrees with only a minor contribution of predominantly short carbon chains < 4 CH$_2$ sequences (McMurry, and Thornton, 1952).